# ATF3 induction prevents precocious activation of skeletal muscle stem cell by regulating H2B expression

Suyang Zhang[1,2], Feng Yang[3], Yile Huang[3], Liangqiang He[2,3], Yuying Li[3], Yi Ching Esther Wan[4,5], Yingzhe Ding[3], Kui Ming Chan [4,5], Ting Xie[6], Hao Sun [3] ✉ & Huating Wang [1,2] ✉

Skeletal muscle stem cells (also called satellite cells, SCs) are important for maintaining muscle tissue homeostasis and damage-induced regeneration. However, it remains poorly understood how SCs enter cell cycle to become activated upon injury. Here we report that AP-1 family member ATF3 (Activating Transcription Factor 3) prevents SC premature activation. *Atf3* is rapidly and transiently induced in SCs upon activation. Short-term deletion of *Atf3* in SCs accelerates acute injury-induced regeneration, however, its long-term deletion exhausts the SC pool and thus impairs muscle regeneration. The *Atf3* loss also provokes SC activation during voluntary exercise and enhances the activation during endurance exercise. Mechanistically, ATF3 directly activates the transcription of *Histone 2B* genes, whose reduction accelerates nucleosome displacement and gene transcription required for SC activation. Finally, the ATF3-dependent H2B expression also prevents genome instability and replicative senescence in SCs. Therefore, this study has revealed a previously unknown mechanism for preserving the SC population by actively suppressing precocious activation, in which ATF3 is a key regulator.

Muscle stem cells (also called satellite cells, SCs) are the adult stem cell population residing in the skeletal muscle, and play a key role in muscle tissue homeostasis and regeneration[1]. SCs normally lie quiescently beneath the muscle fiber between basal lamina and sarcolemma during homeostasis and are marked by expression of the paired box 7 (Pax7) transcription factor[2–4]. Upon muscle damage, SCs can rapidly exit the quiescence, enter the cell cycle to become activated and express myogenic determination factor 1 (MyoD); these activated SCs also known as myoblasts undergo proliferative expansion and differentiate into myotubes, which fuse into myofibers to repair the damaged muscle. Meanwhile, a subpopulation of SCs undergoes a self-renewal

process and returns to the quiescent stage to replenish the SC pool. Each state of the myogenic lineage progression is orchestrated by complex regulatory networks consisting of intrinsic and extrinsic factors/mechanisms[5–9]. Disruption of the regulation may cause SCs to lose the ability to regenerate, contributing to muscle wasting conditions such as muscular dystrophy, cachexia, and sarcopenia. A key advance in our knowledge of SC biology in recent years is the revelation that both quiescence and the transition from quiescence into activation state are highly regulated. For example, upon minor muscle perturbation $G_0$ SCs can enter a $G_{alert}$ or early activating stage which is characterized by enlarged size, enhanced mitochondrial activity and transcriptional

[1]Department of Orthopaedics and Traumatology, Li Ka Shing Institute of Health Sciences, Chinese University of Hong Kong, Hong Kong SAR, China. [2]Center for Neuromusculoskeletal Restorative Medicine, Hong Kong Science Park, New Territories, Hong Kong SAR, China. [3]Department of Chemical Pathology, Li Ka Shing Institute of Health Sciences, Chinese University of Hong Kong, Hong Kong SAR, China. [4]Department of Biomedical Sciences, City University of Hong Kong, Hong Kong SAR, China. [5]Key Laboratory of Biochip Technology, Biotech and Health Centre, Shenzhen Research Institute of City University of Hong Kong, Shenzhen 518172, China. [6]Division of Life Science, The Hong Kong University of Science and Technology, Hong Kong SAR, China. ✉e-mail: haosun@cuhk.edu.hk; huating.wang@cuhk.edu.hk

activation. $G_{alert}$ cells are thus primed for full activation once receiving the external stimuli[10]. Similarly, a report[11] defining the diversity in SCs also classifies quiescent SCs (QSCs) as genuine and primed populations based on CD34 expression level. Understanding the molecular underpinnings of these phases is undoubtedly essential to advance future therapies that can improve regenerative function of SCs, which however is hurdled by the technical difficulty in capturing their resident quiescent state. It is now widely accepted that the isolation processe involving mincing of the muscle, followed by enzymatic digestion and fluorescence-activated cell sorting (FACS) sorting will disturb or destroy the structure and microenvironment of SC niche, which will rapidly induce the adaptive response or early activation of SCs[12-14]. Pre-fixation of the tissue by Paraformaldehyde (PFA) before isolation is believed to preserve the quiescence of SCs to some extent. By comparing the pre-fixed and non-fixed SCs, a number of reports[12-14] converge on the conclusion that SCs undergo massive changes in transcription and in fact enter an early activating stage during the isolation; one of the signature events occurring in the early response to the niche disruption is the rapid induction of Activation Protein-1 (AP-1) family members.

AP-1 transcription factors (TFs) regulate extensive cytological processes, such as proliferation, differentiation, apoptosis, angiogenesis and tumor invasion[15,16]. They are known to function as a dimeric complex that is composed of several monomers from JUN, FOS, ATF, BATF and MAF sub-families, all containing basic regions of bZIP (leucine-zipper) domains[17]. AP-1 family members are very sensitive to a variety of cellular stresses and quickly induced by even slight disturbance[18,19]. In response to the stress, these ATFs display critical roles in the signaling transduction and transcriptional regulation to further regulate cell fate decision, therefore constituting as one set of the most important and well characterized early response genes[20]. Among the AP-1 members, ATF3 belongs to the ATF sub-family and can repress and activate gene transcription, depending on the formation of homo or heterodimers with other ATF/CREB family members[21] and specific binding motifs on the promoter context[22,23]. For example, ATF3 can repress the expression of proinflammatory cytokines induced by the toll-like receptor 4 in the immune response[24]. ATF3 is induced during the early stage of paligenosis to transcriptionally activate the lysosomal trafficking gene Rab7b[25]. Many studies have demonstrated that *Atf3* acts as a stress-inducible gene in rapid response to numerous stress conditions[26-31] and signaling pathways[32-35]. As a key immediate-early response gene with far-reaching impacts, it can act as an integration point for numerous cellular controls, working as a "hub" of the cellular adaptive–response network which is accountable for the disturbance of cellular homeostasis. Nevertheless, the functionality of ATF3 and other AP-1 family members in SCs remains largely unexplored.

In this study, we have delineated the function of ATF3 in SC early activation. It is rapidly and transiently induced upon exposure to external stimuli, such as tissue dissociation, physical exercise, and chemical-induced injury. Short-term deletion of *Atf3* from SCs causes SC activation which accelerates injury-induced regeneration, but long-term *Atf3* deletion leads to the depletion of SC pool and impairs further regeneration. Mechanistically, ATF3 binds to the promoter regions of *Histone2B* (*H2B*) genes to activate their expression; *Atf3* deletion-induced H2B down-regulation accelerates nucleosome displacement during transcription and thus up-regulates the genes involved in SC activation. The H2B loss also results in genome instability and enhanced replicative senescence in SCs. Altogether, our findings have established ATF3 as a key factor for preserving the SC population by suppressing premature activation.

## Results

### ATF3 is rapidly and transiently induced during SC early activation

To explore the transcriptional regulatory events governing the SC quiescence-to-activation transition, we re-analyzed our previously published transcriptomic profiling datasets[36] acquired from SCs in the lineage progression. Briefly, muscles from *Pax7-nGFP* mice[37] were fixed in situ by PFA before subjecting to the standard 3 h long cell dissociation and isolation process, and FACS sorting to obtain QSCs[13], or without prior fixation to obtain freshly isolated early activating SCs (FISC) (Fig. 1a and Supplementary Fig. 1a); FISCs were cultured in vitro for 24, 48 or 72 h to obtain fully activated (ASC-24h), proliferating (ASC-48h) or differentiating (DSC-72h) cells. Consistent with the findings from other recent reports[38-40], AP-1 family genes including *Fos, Atf, Jun* and *Maf* sub-families were dramatically induced in FISCs compared to QSCs (Fig. 1b). Notably, *Atf3* was sharply induced (42.4 fold) in FISCs but was then rapidly decreased at ACS-24h (Fig. 1c, d); this was also confirmed by RT-qPCR (Fig. 1e), indicating that it is a rapid and transient responsive gene to early activation induced by the isolation. Furthermore, immunofluorescence (IF) staining demonstrated the concomitant induction of ATF3 protein in FISCs and a decrease in ASCs (Fig. 1f). Even on the freshly isolated single myofiber that is believed to preserve the quiescent niche to some extent and minimize SC activation[41,42], the associated SCs displayed high expression level of ATF3 protein immediately after 2 h isolation which diminished at 24 h in culture (Fig. 1g), reinforcing the notion that ATF3 is rapidly induced in early activated SCs. To further confirm the ATF3 induction in vivo, BaCl2 was injected into the Tibialis Anterior (TA) muscles of C57BL/6 mice to induce acute muscle damage. In this injury model[36,43,44], tissue degeneration with abundant immune cell infiltrates is normally observed at 1 day post injury (dpi); SCs are rapidly activated to expand as proliferating myoblasts which then fuse to form small new myofibers expressing embryonic myosin heavy chain (eMyHC) and characterized by centrally localized nuclei (CLN); these eMyHC+ fibers are readily seen at 5 dpi. At 7dpi, the muscle is mainly composed of larger regenerated myofibers with down-regulated eMyHC expression; muscle damage and inflammatory cells are largely cleared at 14 dpi while the regenerated myofibers continue to grow in size and mature to achieve full regeneration around 30 dpi. As expected, ATF3 protein was not detected on uninjured TA muscles by IF but readily seen at 1dpi; some staining did not merge with Pax7+ cells, which is in line with its reported ubiquitous induction in multiple cells after injury[45]. At 2dpi ATF3 expression in Pax7+ cells was highly increased, concomitant with the full activation of SCs at this stage (Fig. 1h). Several other AP-1 family members, *Atf4, Fos, FosB,* and *JunB,* shared a similar dynamic expression profile (Fig. 1b–d), which was also confirmed by RT-qPCR (Supplementary Fig. 1b) and immunofluorescent staining (Supplementary Fig. 1c) results. Taken together, our findings show that ATF3 and several other AP-1 family members are rapidly and transiently induced during early SC activation, suggesting their potential roles in the regulation of SC early activation.

### Short-term *Atf3* deletion accelerates acute injury-induced muscle regeneration

To facilitate the dissection of ATF3 function in SCs, we crossed the *Atf3^flox* allele[46], in which two LoxP sites were inserted onto *Atf3* exon2, with *Pax7^CreERT2/CreERT2-R26R^YFP/YFP* mouse[43] to generate control (Ctrl) (*Atf3^+/+- Pax7^CreERT2/CreERT2-R26R^YFP/YFP*) and inducible knockout mice of *Atf3* (*Atf3* iKO) (*Atf3^flox/flox- Pax7^CreERT2/CreERT2-R26R^YFP/YFP*) (Supplementary Fig. 2a). After consecutive 5 days of intraperitoneal (IP) injection of Tamoxifen (TMX) (Fig. 2a), successful deletion of ATF3 in FISCs was confirmed by WB (Supplementary Fig. 2b) or IF (Supplementary Fig. 2c); its depletion was further confirmed by IF staining of SCs on single myofibers (Supplementary Fig. 2d). Morphologically, the 2-month-old iKO showed no apparent difference from the Ctrl littermate mice; no changes in body size and weight were detected either (Supplementary Fig. 2e).

Considering the rapid induction of ATF3 during early SC activation, we reasoned that its loss might impact SC-mediated muscle regeneration. To test this notion, BaCl₂ was injected into the TA

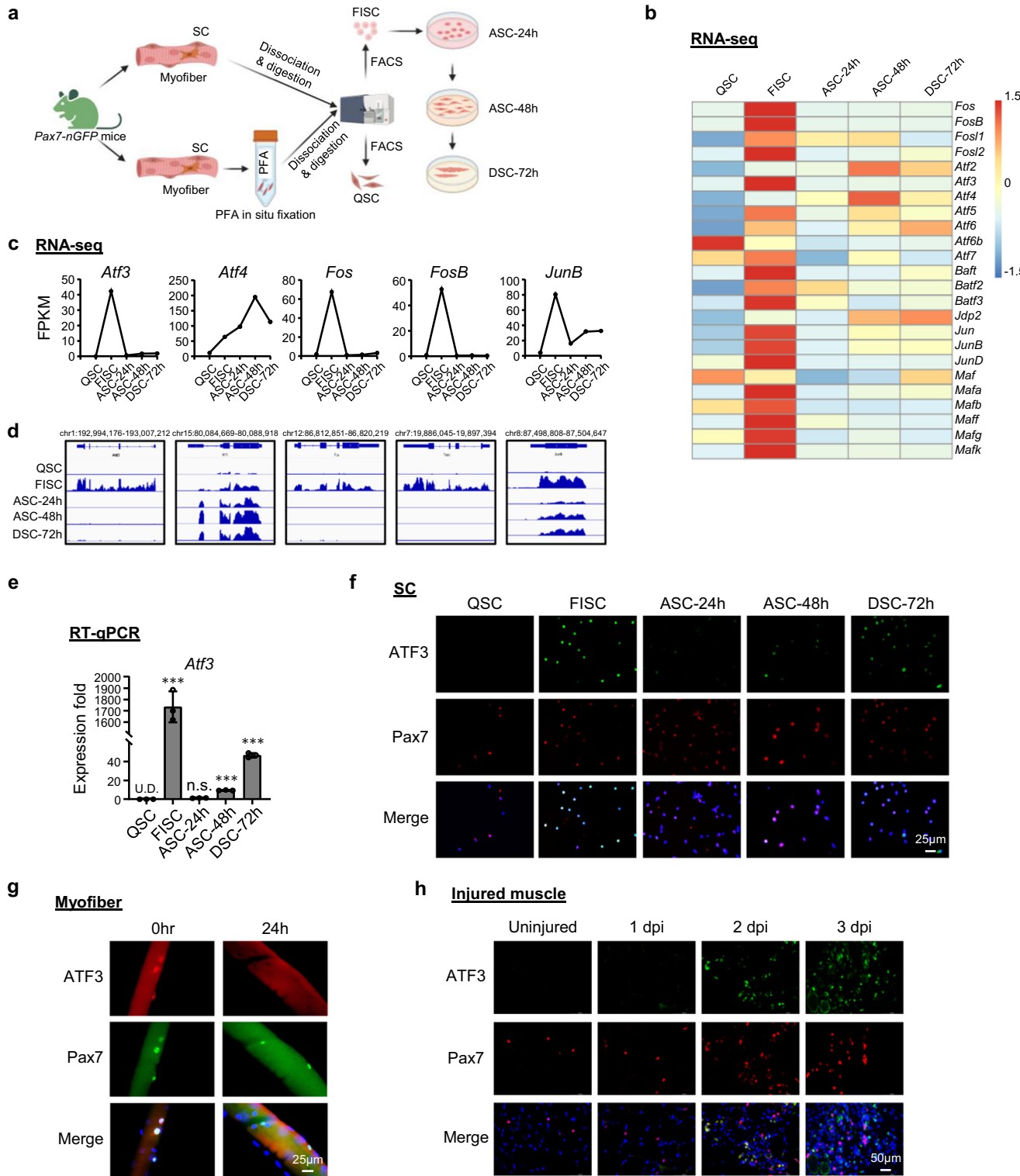

**Fig. 1 | ATF3 is rapidly and transiently induced during SC early activation.**
**a** Schematic for isolation of quiescent satellite cells (QSC) after in situ fixation, freshly isolated SCs (FISC) without prior fixation from muscles of Tg: *Pax7-nGFP* mice. FISCs were subsequently cultured and activated for 24 (ASC-24h), 48 (ASC-48h), or 72 h (DSC-72h). RNAs were extracted for RNA-Seq analysis. Created with BioRender.com. **b** Heat maps indicating gene expression levels (Log2[FPKM]) of AP-1 family TFs detected by the RNA-Seq. **c**, **d** Expression levels (FPKM) and genomic snapshots of *Atf3*, *Atf4*, *Fos*, *FosB*, *JunB* mRNAs from the above RNA-Seq. **e** RT-qPCR detection of *Atf3* in the above cells. *n* = 3 mice per group. *p* = 0.000025,

0.12, 0.0000011 and 0.0000052. **f** Immunofluorescence (IF) staining of ATF3 and Pax7 proteins on the above cells. Scale bar: 50 μm. **g** IF staining of ATF3 and Pax7 proteins on single myofibers from EDL muscles immediately after isolation or cultured for 24 h. Scale bar: 25 μm. **h** IF staining of ATF3 and Pax7 protein on TA muscle sections from uninjured mice or 1, 2 and 3 dpi. Scale bar: 50 μm. All the bar graphs are presented as mean ± SD. Student's *t* test (two-tailed unpaired) was used to calculate the statistical significance (**e**): **p* < 0.05, ***p* < 0.01, ****p* < 0.001. n.s. no significance. Source data are provided as a Source Data file.

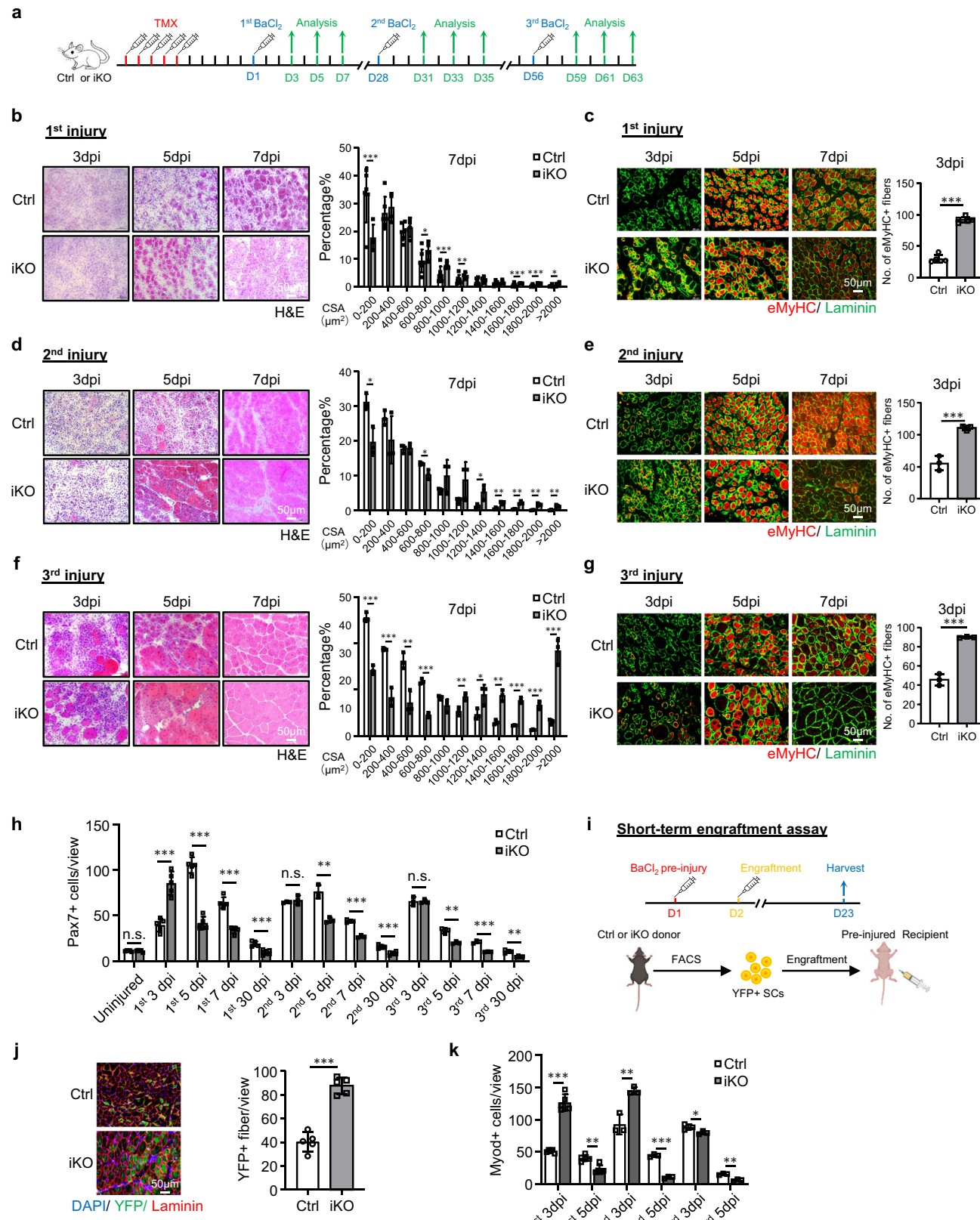

muscles of Ctrl or iKO mice to induce acute muscle damage 5 days after TMX deletion (Fig. 2a). By examining the above injured TA muscles at 3, 5, 7 and 30 dpi (Fig. 2a), the iKO muscles displayed a highly accelerated rate of regeneration. By H&E staining (Fig. 2b), at 5 dpi, regenerating fibers with CLNs were already present in the iKO but not the Ctrl muscles. At 7 dpi the damaged muscle was nearly repaired with

homogenous fibers and no sign of immune cells in the iKO and the fiber size was also significantly larger compared to the Ctrl (the quantification on the right). These experimental observations were further supported by IF staining of eMyHC (Fig. 2c); eMyHC+ fibers were readily observed in the iKO even at 3 dpi (91.64 in iKO vs. 29.86 in Ctrl) but sharply decreased by 7 dpi. Consistently, the number of Pax7+

**Fig. 2 | Short-term *Atf3* deletion accelerates acute injury-induced muscle regeneration. a** Schematic of the experimental design. **b** H&E staining of TA muscles collected at 3, 5 and 7 days post the 1st round injury (dpi). Scale bar: 50 μm. Cross-sectional areas (CSAs) of newly formed fibers were quantified from the TA muscles at 7 dpi. $p = 0.000066$, 0.018, 0.00069, 0.0087, 0.00082, 0.00047 and 0.046. **c** IF staining of eMyHC (red) and Laminin (green) was performed on the above TA muscles. Scale bar: 50 μm. $p = 0.00000011$. **d, e** H&E and eMyHC staining of the TA muscles post the 2nd round injury. Scale bar: 50 μm. $p = 0.012$, 0.019, 0.033, 0.0057, 0.0022, 0.0022 and 0.0093 (**d**); $p = 0.00021$ (**e**). **f, g** H&E and eMyHC staining of the TA muscles post the 3rd round injury. Scale bar: 50 μm. $p = 0.00015$, 0.00087, 0.0063, 0.000074, 0.0089, 0.035, 0.0014, 0.00011, 0.00042 and 0.00071 (**f**); $p = 0.000023$ (**g**). **h** Pax7+ SCs per view were quantified on Ctrl and iKO TA muscles on uninjured or post the 3 rounds of injury. $p = 0.96$, 0.00011, 0.0000013, 0.000014, 0.00066, 0.45, 0.0023, 0.00017, 0.00032, 0.95, 0.0022, 0.00019 and 0.0012. **i** Schematic for engraftment assay. Created with BioRender.com. **j** IF staining of YFP (green) and Laminin (red) on the TA muscles 21 days after engraftment. Scale bar: 50 μm. $n = 5$ mice per group. $p = 0.0000091$. **k** Quantification of Myod+ SCs per view on Ctrl and iKO TA muscles post the 3 rounds of injury. $p = 0.0000012$, 0.0013, 0.0054, 0.000028, 0.041 and 0.0012. $n = 5$ mice per group for 3, 5, 7 days post the 1st injury and 30 days post the 1st, 2nd and 3rd injury and engraftment assay; $n = 3$ mice per group for 3, 5, 7 days post 2nd and 3rd injury. All the bar graphs are presented as mean ± SD, Student's $t$ test (two-tailed unpaired) was used to calculate the statistical significance (**b–h, j, k**): $^*p < 0.05$, $^{**}p < 0.01$, $^{***}p < 0.001$. n.s. no significance. Source data are provided as a Source Data file.

cells at 3 dpi was significantly higher in iKO vs. Ctrl but lower at 5 dpi and 7 dpi (Fig. 2h and Supplementary Fig. 2f). In addition, the TA muscles collected from the iKO mice at 30 dpi had increased muscle weight compared to the Ctrl (43.7 vs. 48.4 mg) (Supplementary Fig. 2g) and enlarged muscle fiber size (1137.4 vs. 862.8 μm²) (Supplementary Fig. 2h), reflecting the enhanced repair.

Surprisingly, the enhanced regenerative ability of *Atf3* iKO SCs persisted after one more round of BaCl₂-induced acute injury given one month later (Fig. 2a). As shown in Fig. 2d, e, the accelerated regeneration remained equally strong based on the increased fiber size at 7 dpi and the number of eMyHC+ fibers at 3 dpi after the 2nd injury. At 30 dpi the iKO muscle also showed increased muscle weight compared to the Ctrl (61.7 vs. 55.8 mg) (Supplementary Fig. 2i) and slightly enlarged muscle fiber size (1724.8 vs. 1633.8 μm²) (Supplementary Fig. 2j). The 3rd round of injury was then given (Fig. 2a) and the expedited repair was still evident (Fig. 2f, g), as the number of eMyHC+ fibers at 3 dpi was still higher in the iKO vs. Ctrl muscles (89.9 vs. 45.7) (Fig. 2g) and a higher number of larger fibers was observed at 7 dpi (Fig. 2f). However, despite slightly increased TA muscle weight in the iKO compared to the Ctrl 30 day post-3rd injury (Supplementary Fig. 2k), there was no significant difference in the average muscle fiber size (Supplementary Fig. 2l), suggesting that the accelerated regeneration in the iKO muscle eventually disappeared after the 3rd injury.

The engraftment assay was then conducted to further validate the enhanced regenerative ability of the iKO SCs. As illustrated in Fig. 2i, 21 days after YFP+ SCs from the Ctrl or iKO mice were injected into the pre-injured TA muscles of receptor nude mice, the TA muscles were collected to evaluate donor cells' regenerative ability based on YFP expression. Indeed, we observed a higher number of YFP+ myofibers in the receptor mice transplanted with the iKO vs. Ctrl SCs (Fig. 2j). Altogether, our results demonstrate that *Atf3* deletion enhances acute injury-induced muscle regeneration.

## *Atf3* deletion provokes premature SC activation and pseudo-regeneration in homeostatic muscle

To dissect the underlying cause for the improved regenerative capacity upon ATF3 loss, we speculated that the iKO SCs underwent rapid activation upon BaCl₂ injury. Indeed, when plotting the dynamics of MyoD+ SCs along the course of the three rounds of injury/regeneration. a much higher number of MyoD+ cells were detected at 3 dpi (Fig. 2k and Supplementary Fig. 2m), but the cell number quickly declined at 5 dpi. Such phenomenon existed during the first two rounds of injuries but disappeared in the third round (Fig. 2k and Supplementary Fig. 2m). Compared to the Ctrl, the faster activation of iKO cells upon acute injury implies the quicker transition from quiescence to activation. Indeed, FISCs from the *Atf3*-iKO mice displayed a strikingly increased propensity for cell cycle entry based on EdU incorporation 24 h after culture (~23% increase) (Fig. 3a). In addition, Pax7+MyoD+ cells in the iKO exhibited 10% increase compared to the Ctrl 24 h after culture (Fig. 3b).

Furthermore, the FISCs from the iKO muscles were also slightly larger in size than the Ctrl ones (86.1 vs. 72.9) (Fig. 3c). Consistently, isolated single myofibers from the iKO muscles had significantly higher EdU incorporation (14% increase) than the Ctrl ones 36 h after culture (Fig. 3d), reinforcing the faster cell cycle entry of the iKO SCs. The iKO SCs on freshly isolated single myofibers also produced 1.90-fold and 2.08-fold more Pax7+MyoD+ cells than those Ctrl SCs without culturing or with 3 h (Fig. 3e, f). Altogether, these results indicate that the loss of ATF3 results in the release from quiescence and thus rapid SC activation without acute injury.

The enhanced regenerative ability of iKO SCs prompted us to further test if the ATF3 loss may lead to spontaneous activation and pseudo-regeneration in muscles without injury. Compared to the Ctrl, there was no increase of Pax7+ SCs in the iKO TA muscles at 5 days post-TMX injection but a significant increase of Pax7+ cells in the iKO TA muscles at 21 (13.4 vs. 9.58) and 28 days (11.7 vs. 8.9) (Figs. 3g and S3a), indicating the occurrence of spontaneous activation and expansion of iKO SCs in homeostatic muscles. Interestingly, by 56 days post-TMX injection, the number of Pax7+ cells in the iKO muscles finally declined compared to the Ctrl (Figs. 3g and S3a); this was accompanied by readily seen CLN+ fibers (Fig. 3h), indicating the fusion of the activated SC-derived myoblasts to existing fibers, which is known as pseudo-regeneration[47]. To confirm these findings, we performed five doses of EdU injections at 5 days post-TMX injection to show that there were more EdU+ cells on the muscle sections of the iKO than the Ctrl at 21 days post-EdU injection (Fig. 3i). Similarly, with a single dose of EdU administered, EdU+ cells were only detected on the isolated iKO myofibers or TA muscles 12 h later but not on the Ctrl ones (Supplementary Fig. 3b). Altogether, these results suggest that *Atf3* deletion provokes the precocious activation of SCs in homeostatic muscles thus ATF3 may function to actively suppress SC activation. This was further strengthened by overexpressing ATF3 in FISCs by a lentivirus (Supplementary Fig. 3c); it significantly inhibited SC activation (Fig. 4a, 20.9% reduction of Pax7+MyoD+ cells and Fig. 4b, 21.3% reduction of EdU+ cells). Moreover, overexpressing ATF3 in vivo by intramuscular injection of the lentivirus particles at 1 dpi (Fig. 4c) delayed the regeneration in Ctrl mice and also blunted the accelerated regeneration in iKO mice (Figs. 4d–h and S3d, e).

In addition, we observed there was 24% increase of MyoD+ MyoG+ differentiating cells in the isolated iKO muscle fibers compared to the Ctrl after 72 h culture (Supplementary Fig. 3f), suggesting that *Atf3* deletion may accelerate SC progeny differentiation. This was confirmed by staining for MyoD and MyoG on SCs cultured for 72 h; 9% increase in the double positive cells was detected in the iKO vs. Ctrl (Supplementary Fig. 3g). Interestingly, there was also a slight decrease of Pax7+MyoD− cells in the iKO muscle fibers compared to the Ctrl (10.13% vs. 8.48%) (Supplementary Fig. 3h), suggesting that Atf3 loss may also lead to impaired self-renewal. Consistently, we also noticed a progressive decline in SC number after each round of injury (Fig. 2h and Supplementary Fig. 2f).

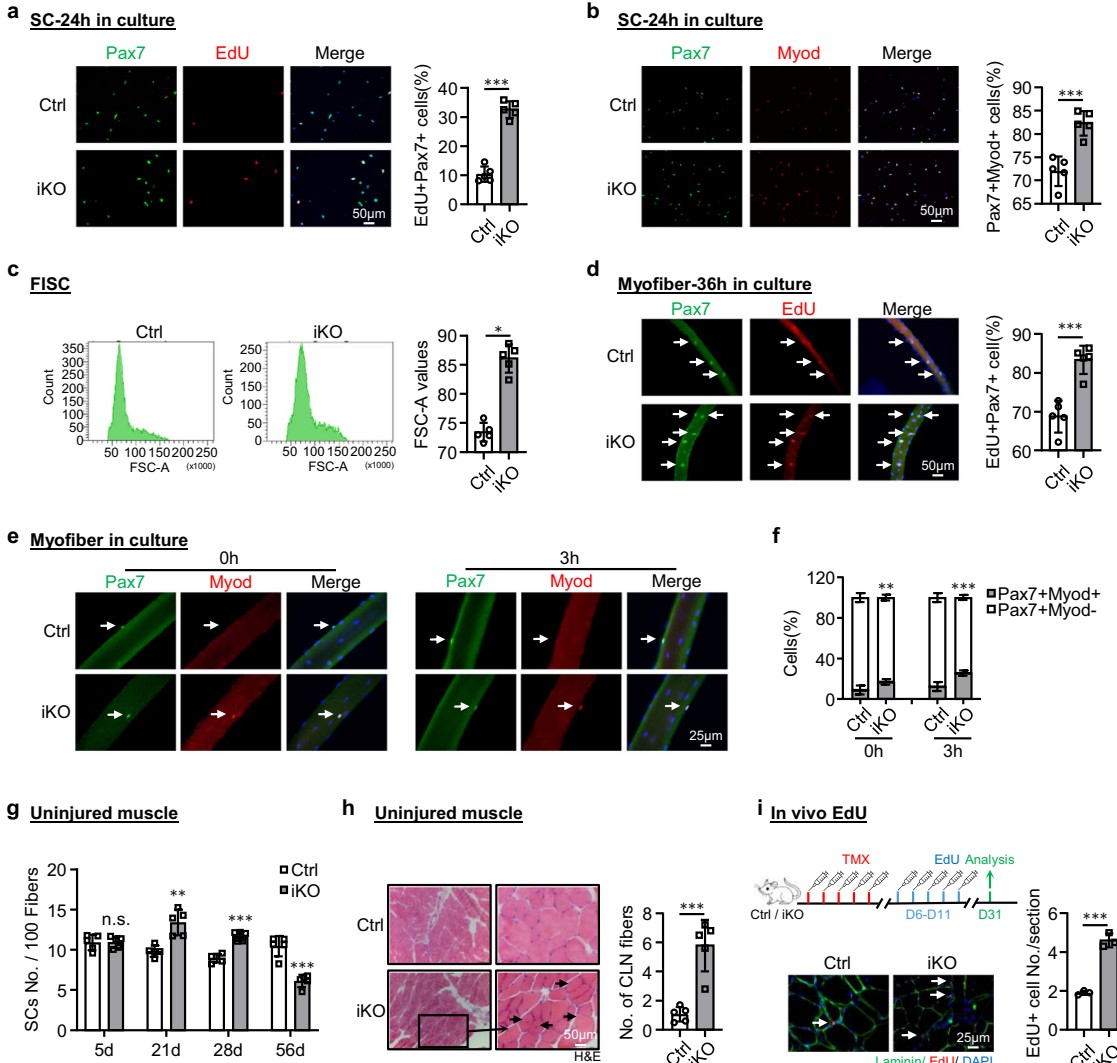

**Fig. 3 | *Atf3* deletion provokes premature SC activation and pseudo-regeneration in homeostatic muscle. a** SCs from Ctrl or iKO mice were cultured for 24 h and treated with EdU for 6 h before staining for EdU (red) and Pax7 (green). Scale bar: 50 μm. *p* = 0.0000014. **b** IF staining of Pax7 (green) and MyoD (red) on the above SCs cultured for 24 h. Scale bar: 50 μm. *p* = 0.00050. **c** Representative FACS plots showing the size of FISCs sorted from Ctrl or iKO muscles. *p* = 0.050. **d** Freshly isolated myofibers from Ctrl or iKO mice were cultured for 36 h and treated with EdU for 6 h before staining for EdU (red) and Pax7 (green). Scale bar: 50 μm. *p* = 0.00034. **e** IF staining of Pax7 (green) and MyoD (red) on the above myofibers immediately after isolation or cultured for 3 h. Scale bar: 25 μm. **f** Quantification of the percentage of Myod-Pax7+ and Myod+Pax7+ SCs. *p* = 0.0098

and 0.00035. **g** Quantification of the numbers of Pax7+ SCs per 100 fibers on uninjured Ctrl or iKO muscles 5, 21, 28 and 56 days after TMX injection. *p* = 0.96, 0.0018, 0.000049 and 0.00018. **h** H&E staining of the above uninjured muscles 56 days after TMX injection. Scale bar: 50 μm. *p* = 0.00043. **i** Upper: Schematic outline of the in vivo EdU assay performed on uninjured Ctrl or iKO muscle; EdU was injected by IP 5 days after TMX. The muscles were collected 21 days later. Lower IF staining of EdU (red) and Laminin (green). Scale bar: 25 μm. *p* = 0.00024. *n* = 5 mice per group (**a**–**h**); *n* = 3 mice per group (**i**). All the bar graphs are presented as mean ± SD, Student's *t* test (two-tailed unpaired) was used to calculate the statistical significance (**a**–**d**, **f**–**i**): \**p* < 0.05, \*\**p* < 0.01, \*\*\**p* < 0.001. n.s. no significance. Source data are provided as a Source Data file.

## Long-term *Atf3* deficiency depletes SC pool and impairs muscle regeneration

Next, to test if long-term *Atf3* deficiency causes SC exhaustion and impairs muscle regeneration, we induced muscle injury by BaCl₂ administration 30 days or 120 days after deleting *Atf3* from SCs and examined the injured TA muscles at 3, 5 and 7 dpi (Fig. 5a). Thirty days after the deletion, the muscle regeneration (based on the muscle fiber size and the eMyHC+ fiber number) showed no significant changes compared to the Ctrl mice (Fig. 5b, c). The number of Pax7+ SC was also comparable at 3 and 7 dpi and only slightly lower in the iKO at 5 dpi (Fig. 5d and Supplementary Fig. 4a). In contrast, 120 days after the deletion, the muscle regeneration was obviously compromised in the iKO compared to the Ctrl mice. A significant increase of smaller fibers was observed at 7 dpi (Fig. 5e); the number of eMyHC+ fibers was much lower at 5 dpi but higher at 7 dpi (Fig. 5f); the number of Pax7+ SCs was

also significantly reduced at both 3 and 7 dpi (Fig. 5g and Supplementary Fig. 4b). These results demonstrate that the long-term *Atf3* deficiency is detrimental to regeneration by shrinking the SC pool. Interestingly, when the same number of YFP+ cells isolated from the Ctrl or iKO mice 4 months after TMX injection were used in the engraftment assay (Fig. 5h), a higher number of YFP+ myofibers were observed in the recipient nude mice transplanted with the iKO cells (59.28 vs. 31.72, Fig. 5i), suggesting that enhanced regenerative ability of the iKO cells persists after long term ATF3 loss and the impaired muscle regeneration indeed arises from the reduced cell pool.

To further establish the importance of *Atf3* in SC establishment and maintenance, we successfully deleted *Atf3* in Pax7+ myogenic progenitors by crossing the *Atf3^flox* allele with a non-inducible *Pax7^Cre*-*R26R^YFP* transgenic mouse strain[48], in which the Cre recombinase is expressed in Pax7+ progenitor cells in as early as E9.5 (Supplementary

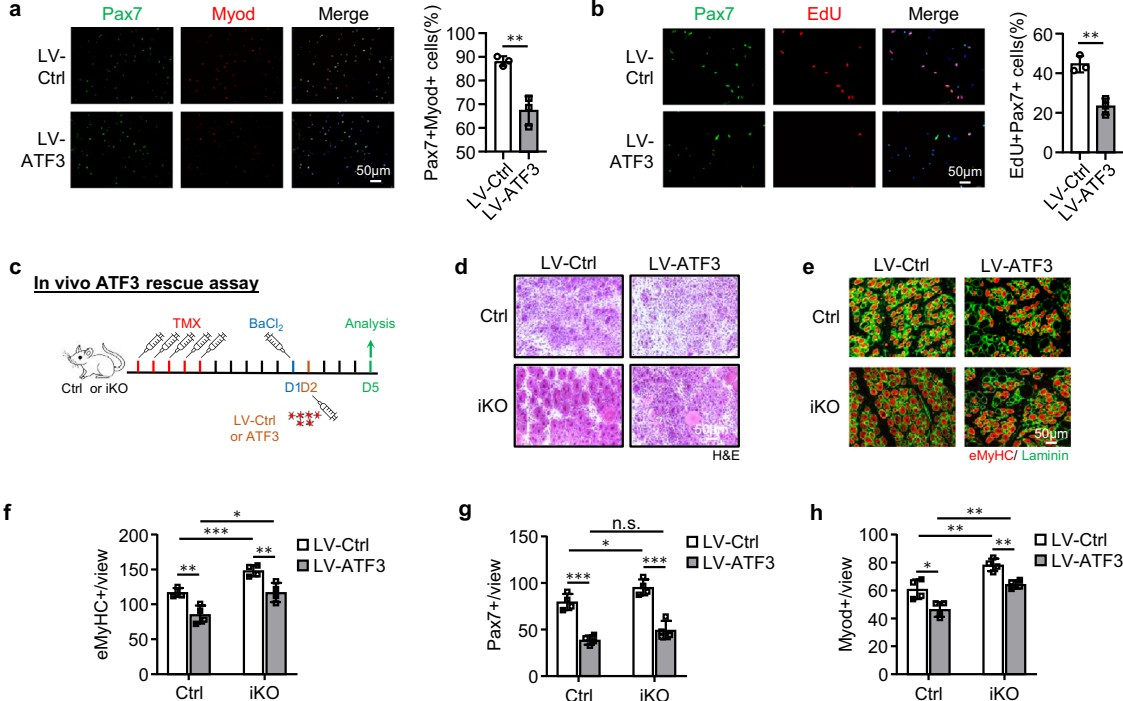

**Fig. 4 | ATF3 re-expression in iKO reverts the accelerated SC activation and muscle regeneration. a** FISCs from Ctrl or iKO mice were infected with ATF3-overexpressing lentiviruses and stained for Pax7 (green) and Myod (red) after 48 h. Scale bar: 50 μm. *p* = 0.0050. **b** EdU assay was performed and EdU+ cells were quantified. Scale bar: 50 μm. *p* = 0.0031. **c** Schematic for overexpressing ATF3 in vivo. ATF3 expressing lentiviruses were injected into Ctrl or iKO TA muscles 1 day after BaCl₂ injury. The muscles were collected 5 days after infection for analysis. **d** H&E staining of the above TA muscles. Scale bar: 50 μm. **e** IF staining of eMyHC (red) and Laminin (green) was performed on the above TA muscles. Scale bar:

50 μm. **f** The numbers of eMyHC+ fibers per view were quantified. *p* = 0.0044, 0.00081, 0.015 and 0.0072. **g** The numbers of Pax7+ cells per view on the above TA muscles were quantified. *p* = 0.00018, 0.041, 0.098 and 0.00040. **h** The numbers of Myod+ cells per view on the above TA muscles were quantified. *p* = 0.016, 0.0058, 0.0010 and 0.0019. *n* = 3 mice per group (**a**, **b**); *n* = 4 mice per group (**d**–**h**). All the bar graphs are presented as mean ± SD. Student's *t* test (two-tailed unpaired) was used to calculate the statistical significance (**a**, **b**, **f**–**h**): \**p* < 0.05, \*\**p* < 0.01, \*\*\**p* < 0.001. n.s. no significance. Source data are provided as a Source Data file.

Fig. 5a–c). Interestingly, the cKO muscles exhibited 84% increase of Pax7+ SCs compared to the Ctrl around 1 month after birth when SC pool was established[49], but 79% decrease in two-month-old adult mice (Supplementary Fig. 5e), indicating that ATF3 is required for SC maintenance at adulthood but dispensable for SC establishment. A decreased TA muscle weight was detected in cKO at 1-month-old (29.42 mg vs. 34.88 mg, Supplementary Fig. 5f) but no significant difference in the body weight (12.66 mg vs. 12.42 mg, Supplementary Fig. 5d) and fiber size (861.96 μm² vs. 848.71 μm², Supplementary Fig. 5g), it is thus hard to conclude whether muscle hypertrophy occurs in the young cKO mice. Expectedly, when the adult SCs were isolated and cultured, cKO cells displayed a much higher propensity for activation assessed by both Pax7+Myod+ staining (Supplementary Fig. 5h, 23.8% increase in cKO vs. Ctrl) and EdU assay (Supplementary Fig. 5i, 99.9% increase). Correspondingly, the regenerative capacity after acute injury was evidently compromised in cKO muscles (Supplementary Fig. 5j–l). Therefore, the loss of ATF3 induces precocious SC activation and leads to the eventual reduction of the SC pool.

## *Atf3* deletion enhances SC activation during voluntary and endurance exercises

Short term and non-strenuous voluntary exercise (VE) fails to cause SC activation but endurance exercise (EE) activates SCs[50–54]. To further prove the function of ATF3 in preventing SC activation, we sought to determine if the ATF3 depletion influences the VE-induced SC activation. Immediately after five doses of TMX injection, Ctrl and iKO mice were subject to an established voluntary wheel running regime; the mice were provided with access to free rotating running wheels; a stable VE routine was reached within 7 days of training and continued

for another 21 days long VE (Fig. 6a). The daily running distance was recorded and no significant difference was detected between Ctrl (14.38 km) vs. iKO (13.78 km) mice (Supplementary Fig. 6a, b).

Expectedly, the expressions of AP-1 family members were induced by the VE in Ctrl SCs (Supplementary Fig. 6c). Also consistent with prior reports[54], the VE did not induce SC activation in the Ctrl mice as no MyoD+ cells were detected on the muscle sections before and after the VE (Fig. 6b and Supplementary Fig. 6d); consistently, Pax7+MyoD+ staining of FISCs (Fig. 6c) or EdU staining of ASCs cultured for 24 h (Fig. 6d) did not reveal increased activation rate before and after the VE. In contrast, the iKO cells were readily activated by the VE (Fig. 6b–d and Supplementary Fig. 6d). Consistently, in the iKO muscles, we also detected a significantly increased number of CLN fibers (136% increase) (Fig. 6e) and evident hypertrophy (fiber size increased by 43.3%) after the VE (Fig. 6f). In the Ctrl muscles, the number of CLN was increased after the VE but not enough to cause hypertrophic growth (Fig. 6e, f). The above findings thus reinforce the notion that ATF3 loss induces rapid activation of SCs by the VE. As a result, there was reduced SC pool (38.2 % decrease of Pax7+ FISCs) at the end of the VE regime (Fig. 6g and Supplementary Fig. 6e).

Since EE is known to cause SC activation and muscle hypertrophy[50–53], we next examined if the *Atf3* deficiency affects the EE-induced SC activation. To this end, Ctrl and iKO mice were subject to a treadmill running regime[55,56] in which the treadmill was set at a 5° incline and a speed of 20 cm/s for 60 min. After five doses of TMX injection, the mice were trained for a 5-day adaption period, then followed by a 10-day endurance training (+EE) or, as the control condition, without any training (NE) (Fig. 6h). As expected, the expressions of AP-1 family members were rapidly induced by the EE in SCs

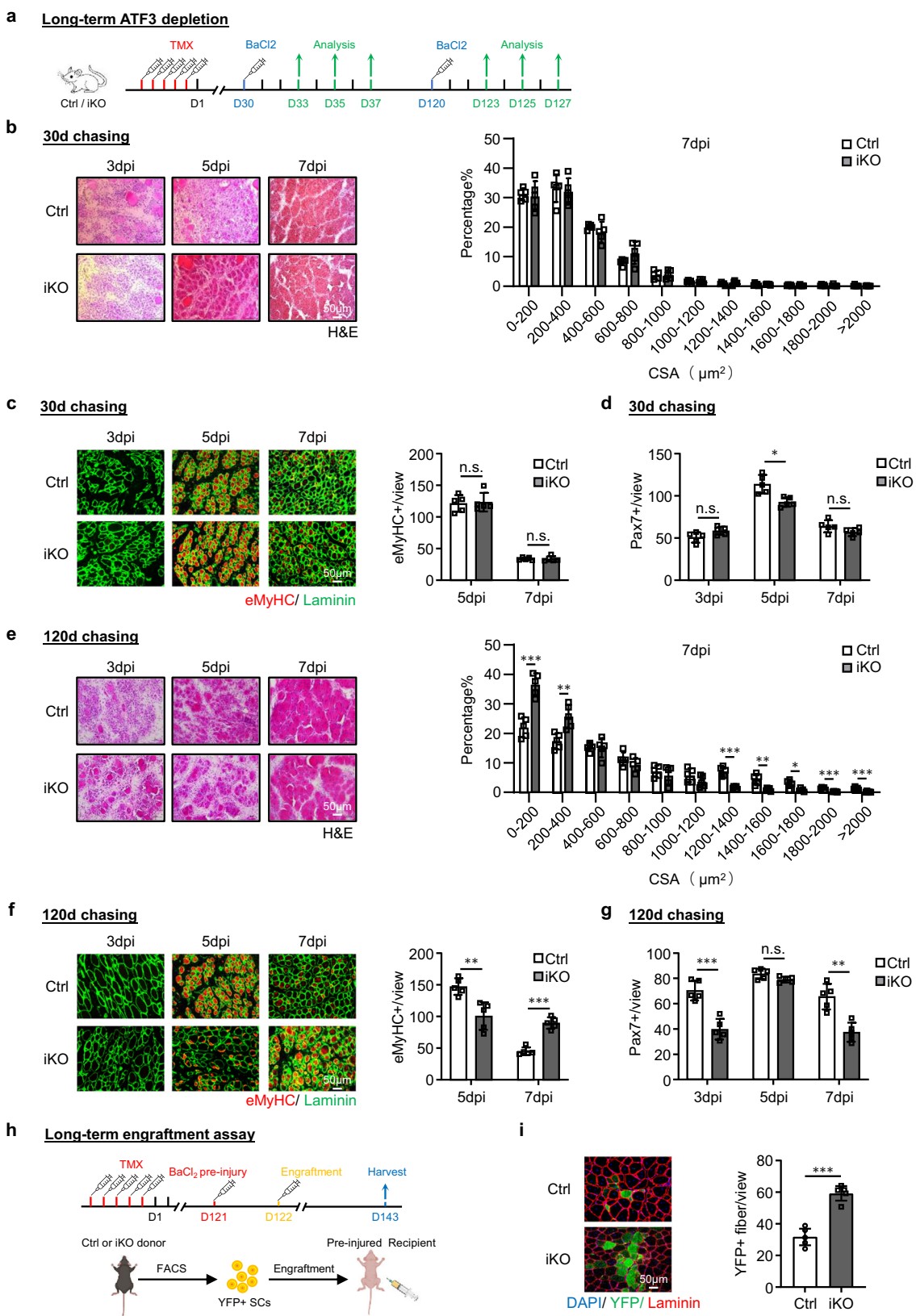

(Supplementary Fig. 6c). Furthermore, the EE caused evident SC activation in the Ctrl muscles, but the activation was much stronger in the iKO muscles (Fig. 6i–k and Supplementary Fig. 6f). Consistently, CLN fibers were induced in both Ctrl and iKO muscles after the EE; and a higher increase of CLN fibers was seen in the iKO compared to the Ctrl muscles (Fig. 6l). In addition, both Ctrl and iKO muscles showed

hypertrophic growth after the EE but no difference was detected (Fig. 6m). At the end of the EE regime, the number of Pax7+ FISCs was significantly increased in the Ctrl but decreased in the iKO (Fig. 6n and Supplementary Fig. 6g), indicating the rapid exhaustion of the iKO SC pool. Altogether, these findings indicate that the loss of ATF3 provokes SC activation by VE and also enhances SC activation by EE.

**Fig. 5 | Long-term *Atf3* deficiency depletes SC pool and impairs muscle regeneration. a** Schematic outline of the experimental design for testing the effect of long-term *Atf3* deletion on muscle regeneration process. 30 or 120 days chasing after TMX injection was given before BaCl$_2$ injection. **b** Left: H&E staining of the TA muscles collected at 3, 5 and 7 dpi after the 30 days chasing period. Scale bar: 50 μm. *n* = 5 mice per group. Right: CSAs of newly formed fibers were quantified from the above-stained TA muscle at 7 dpi and the distribution is shown. *n* = 5 mice per group. **c** Left: IF staining of eMyHC (red) and Laminin (green) was performed on the above TA muscles. Scale bar: 50 μm. Right: Quantification of the numbers of eMyHC+ fibers per view at 5 and 7 dpi. *n* = 5 mice per group. *p* = 0.83 and 0.98. **d** Quantification of the numbers of Pax7+ SCs per view. *n* = 5 mice per group. *p* = 0.054, 0.0047 and 0.11. **e–g** The above assays/quantifications were performed on the Ctrl or iKO muscles collected at 3, 5 and 7 dpi after the 120 days chasing period. Scale bar: 50 μm. *n* = 5 mice per group. *p* = 0.00020, 0.0047, 0.000068, 0.0024, 0.013, 0.00019 and 0.00057 (**e**); *p* = 0.0034 and 0.000018 (**f**); *p* = 0.00031, 0.063 and 0.0010 (**g**). **h** Schematic for the engraftment assay. 120 days after TMX injection, FISCs from donor mice (Ctrl/iKO) were injected into pre-injured recipient nude mice. TA muscles were collected 21 days after engraftment for analysis. Created with BioRender.com. **i** Left: IF staining of YFP (green) and Laminin (red) on the above TA muscles. Scale bar: 50 μm. Right: the numbers of YFP+ fibers per view were quantified. *n* = 5 mice per group. *p* = 0.000047. All the bar graphs are presented as mean ± SD. Student's *t* test (two-tailed unpaired) was used to calculate the statistical significance (**b–g, i**): *\*p* < 0.05, *\*\*p* < 0.01, *\*\*\*p* < 0.001. n.s. no significance. Source data are provided as a Source Data file.

## ATF3 regulates *H2B* gene expression and nucleosome patterning

Altogether the above results suggest an essential role of ATF3 induction in preventing precocious SC activation, to illuminate the underlying molecular mechanism, we profiled the ATF3 transcriptional output by performing RNA-Seq in FISCs collected from the Ctrl and iKO muscles (Fig. 7a). A total of 1866 transcripts (red dots) (79.9%) were up-regulated in the iKO compared to the Ctrl, while 469 (blue dots) (20.1%) were down-regulated (Fig. 7b, Supplementary Fig. 7a and Supplementary Data 1), suggesting that the ATF3 loss induced global transcriptional activation. The GO analysis revealed that the up-regulated genes were enriched for "mitochondrion" "extracellular matrix", etc. (Fig. 7c and Supplementary Data 1). Of note, elevated mitochondrial activity and extracellular matrix expression have previously been described in the G$_{Alert}$ cells[10], suggesting that ATF3 functions to repress the expression of these activating genes, thus preventing premature SC activation. The down-regulated genes were interestingly enriched for "nucleosome" and "nucleosome assembly", etc. (Fig. 7d and Supplementary Data 1). Notably, the genes encoding Histone proteins were highly represented among the down-regulated genes (Fig. 7e), which was also confirmed by RT-qPCR (19–46% decrease) (Supplementary Fig. 7b). Histone encoding genes are typically organized into multigene clusters and H2B protein is encoded by 2 gene clusters with 15 on Chr13 forming a *Hist1h2b* cluster and 2 on Chr3 forming a *Hist2h2b* cluster (Supplementary Fig. 7c). Remarkably, 13 out of the 17 *H2b* genes were down-regulated in the iKO SCs (Fig. 7e). Additionally, we also performed RNA-Seq on FISCs isolated from the Ctrl and *Atf3*-cKO mice and similar results were obtained (Supplementary Fig. 7d–g and Supplementary Data 3). These results demonstrate that the ATF3 loss decreases *H2b* gene expression.

Next, we performed ChIP-Seq on the C2C12 myoblasts with exogenous ATF3 over-expression as a surrogate system to define direct binding targets of ATF3 (Fig. 7a and Supplementary Fig. 7h). A total of 2871 binding peaks were identified in 869 genes with 60%, 28% and 9% located in intergenic regions, introns and promoters, respectively (Fig. 7f and Supplementary Data 2). These peaks were enriched for the known ATF3 binding motif, TGACTCA (Fig. 7g), attesting to the good data quality. GO analysis of these ATF3 bound genes indicated remarkable enrichment for the terms related to nucleosome (Fig. 7h and Supplementary Data 2). By integrating the RNA-Seq and ChIP-Seq results, 72 down-regulated genes contained ATF3 binding sites and again they were enriched for nucleosome related GO terms and many were histone genes (Fig. 7i and Supplementary Data 2). Notably, 8 of the 13 down-regulated *H2b* genes had ATF3 binding sites in their promoters (Fig. 7i, j, Supplementary Fig. 7i and Supplementary Data 2). Meanwhile, 112 up-regulated genes possessed ATF3 binding and were enriched for mitochondrion related GO terms (Supplementary Fig. 7j and Supplementary Data 2). The Western blot (Fig. 7k) and IF staining (Fig. 7l) results confirmed a substantial reduction or near loss of H2B protein in the iKO FISCs while other histone proteins such as H3, H4 and H2A remained unaltered. These results demonstrate that *H2b* genes are bona fide transcriptional targets of ATF3.

Global diminution of histones leads to decreased amount of nucleosomes, augmented nucleosome spacing, and alters nucleosome occupancy thus affecting transcription[57–59], to determine if the H2B loss in the iKO SCs alters nucleosome positioning and occupancy, CUT&RUN assay[60,61] was performed to map the genomic localization of H2B in Ctrl and iKO cells (Fig. 7a). A total of 272590 bins were defined genome-wide from the three replicates (Supplementary Data 4). Unexpectedly, the average H2B CUT&RUN signals were largely unaltered in the iKO vs. Ctrl with only 3429 bins (1.4%) showing changes (Fig. 7m, n). Nevertheless, 3220 out of the altered bins indeed showed decreased H2B enrichment in various genomic regions, including promoter (12%), gene body (39%) and intergenic regions (49%) (Fig. 7m). Since the nucleosome is partly disassembled through the removal of one H2A-H2B dimer to facilitate Pol II transcription[61] (Fig. 7o), we hypothesized that the local loss of H2B could facilitate nucleosome destabilization to promote the transcription of associated genes. Indeed, by intersecting with RNA-Seq data, 137 up-regulated genes showed decreased H2B signals on their promoters or gene bodies (Fig. 7p, q and Supplementary Data 4). These H2B target genes were enriched for GO terms related to muscle differentiation and also included *Id3*, *IGF-1* and *Jun*, etc. (Fig. 7r, s), which was consistent with the activation state of the iKO SCs[11,62,63]. Taken together, these findings demonstrate that the decreased H2B level could mediate the precocious SC activation following the ATF3 loss.

## H2B mediates ATF3 function in SC activation and muscle regeneration

To solidify the functional link between ATF3 and H2B in SC activation and muscle regeneration, we found that overexpressing H2B by transfecting a pcDNA-H2B plasmid in FISCs indeed repressed the accelerated activation of Atf3 iKO cells but no impact on Ctrl cells (Fig. 8a, b). And in vivo, we also overexpressed H2B (Supplementary Fig. 7k) by intramuscular injection of lentiviral particles at 1dpi (Fig. 8a). As shown in Fig. 8c–g, this also restored the acute injury induced regeneration in the Atf3-iKO mice: a significantly lower number of eMyHC+ fibers were observed 5 dpi (Fig. 8d, e), which was accompanied by reduced numbers of Pax7+ cells (Fig. 8f) and MyoD+ cells (Fig. 8g). Altogether these findings validate that H2B loss mediates the precocious SC activation and enhanced muscle regeneration observed upon ATF3 loss.

## H2B loss increases DNA damage and senescence in *Atf3* iKO SCs

To further elucidate the consequence of H2B loss in SCs, we examined genome instability and cellular senescence considering histone proteins are important for chromatin integrity and genome stability[57,59]. FISCs from iKO or Ctrl muscles were treated with various dosages (0, 1000 or 2000J) of UV light and comet assay was performed. Indeed, a substantial increase in the comet tail length was observed under all three dosages in the iKO as compared to the Ctrl (Fig. 9a), suggesting that H2B decrease causes genomic instability. This was further confirmed by a much higher percentage of γH2AX+ cells in the cultured iKO cells compared to the Ctrl (Fig. 9b, c). To examine if the genomic

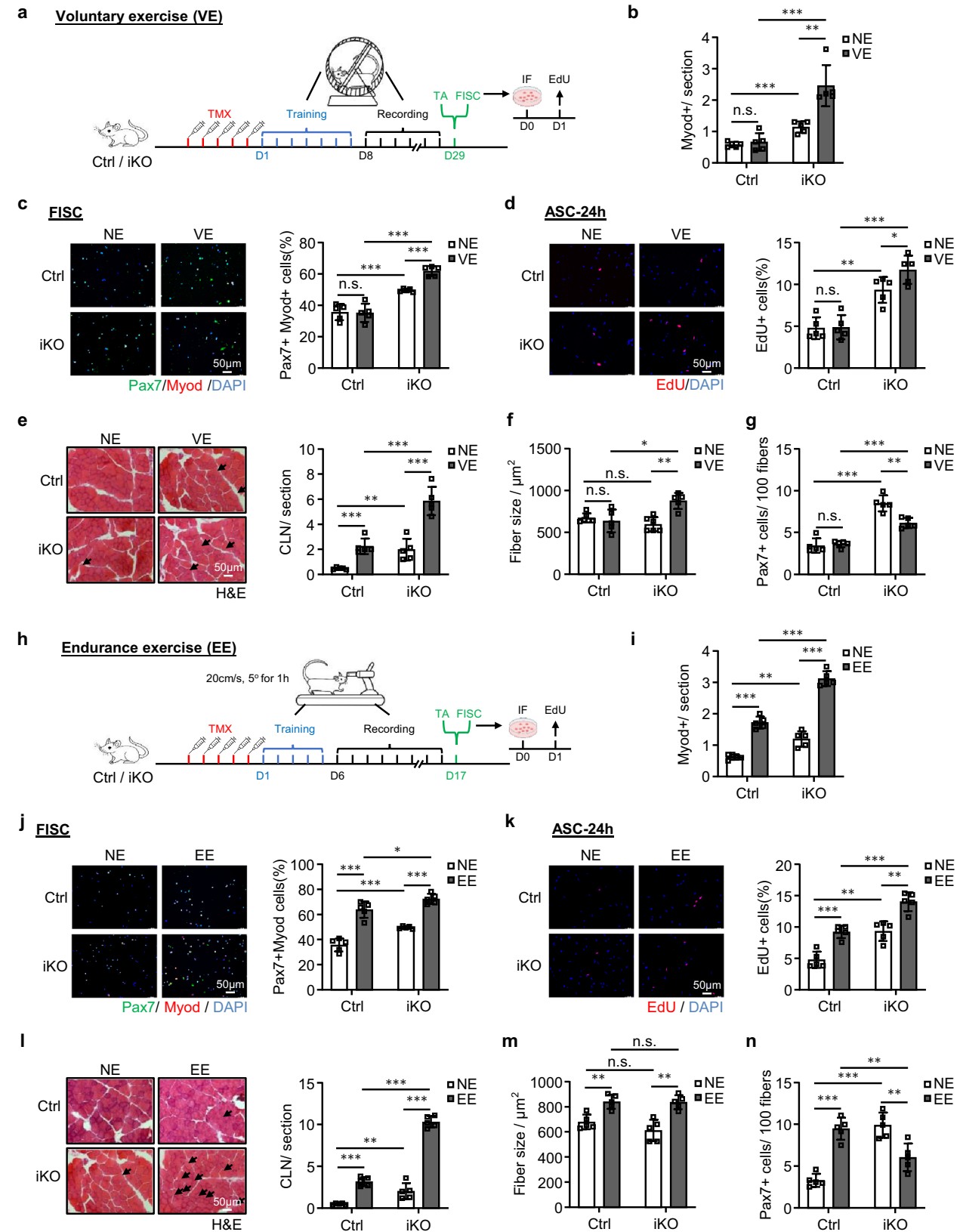

instability leads to cellular senescence, cells were cultured in growth medium for 9 days and β-Gal staining was performed; a 19% increase of β-Gal+ cells was detected in the iKO compared to the Ctrl (Fig. 9d). Consistently, a significant up-regulation of *p16*, *p21*, and *p53* mRNAs (Fig. 9e) and a higher level of p21 and p53 proteins were also detected in the iKO (Fig. 9f). Altogether, the above findings demonstrate that

H2B loss leads to increased genome instability and cellular senescence in the *Atf3* iKO cells. Expectedly, H2B over-expression in the iKO cells reduced the DNA damage accumulation and inhibited cellular senescence as assessed by decreased p21 protein (Fig. 9g) as well as *p16*, *p21* and *p53* mRNAs (Fig. 9h). These results indicate that the decreased H2B expression causes SCs to undergo replicative senescence.

**Fig. 6 | *Atf3* deletion induces SC activation during voluntary and endurance exercise. a** Schematic of the experimental design for VE. **b** Quantification of Myod+ cells on the above TA muscles. *p* = 0.54, 0.00024, 0.00049 and 0.0024. **c** IF staining of Pax7 (green) and MyoD (red) on SCs isolated from the above mice and cultured for 24 h sorted. Scale bar: 50 μm. *p* = 0.91, 0.00035, 0.000018 and 0.000032. **d** EdU (red) was stained on SCs sorted from the above mice and cultured for 24 h. Scale bar: 50 μm. *p* = 0.90, 0.0010, 0.00013 and 0.046. **e** H&E staining of the above TA muscles. Scale bar: 50 μm. *p* = 0.00043, 0.0073, 0.00021 and 0.00047. **f** Fiber size of the above TA muscles (**e**). *p* = 0.54, 0.19, 0.012 and 0.0017. **g** Quantification of Pax7+ SCs on the above TA muscles. *p* = 0.66, 0.000023, 0.000052 and 0.0017. **h** Schematic of the experimental design (EE). **i** Quantification of Myod+ cells on the TA muscles collected from the above Ctrl and iKO mice with or without EE.

*p* = 0.0000026, 0.0010, 0.0000073 and 0.0000015. **j** IF staining of Pax7 (green) and MyoD (red) on SCs sorted from the above mice and cultured for 24 h. Scale bar: 50 μm. *p* = 0.000062, 0.00035, 0.035 and 0.0000018. **k** EdU (red) was stained on SCs isolated from the above mice and cultured for 24 h. Scale bar: 50 μm. *p* = 0.00031, 0.0010, 0.00034 and 0.0012. **l** H&E staining of the above TA muscles. Scale bar: 50 μm. *p* = 0.0000039, 0.0073, 0.000000071 and 0.00000026. **m** Fiber size of the above TA muscles (**l**). *p* = 0.0022, 0.19, 0.91 and 0.0012. **n** Quantification of Pax7+ SCs per on the above TA muscles. *p* = 0.000018, 0.000024, 0.0067 and 0.0049. *n* = 5 mice per group (**b–g**, **i–n**). All the bar graphs are presented as mean ± SD. Student's *t* test (two-tailed unpaired) was used to calculate the statistical significance (**b–g**, **i–n**): \**p* < 0.05, \*\**p* < 0.01, \*\*\**p* < 0.001. n.s. no significance. Source data are provided as a Source Data file.

## Screening of other functional AP-1 family members in SCs and regeneration

After the above holistic characterization of the function and mechanism of ATF3 in SCs, we sought to expand the study by asking if other AP-1 family TFs such as ATF4, FOS, FOSB, and JUNB, also play functional roles in SCs considering their similar rapid and transient induction dynamics during SC early activation (Figs. 1b–d and S1c). To test the idea, we conducted a functional screening using our recently developed *Pax7*[Cas9]/AAV-sgRNA mediated in vivo genome editing platform[64], which is based on a Cre-dependent Cas9 knock-in mice and AAV-mediated sgRNAs delivery. Briefly, one pair of sgRNAs targeting each TF were selected and packaged into AAV9 virus particles; $4 \times 10^{11}$ viral genomes (vg)/mouse of AAV9-sgRNA was intramuscularly (IM) injected into the TA muscles of *Pax7*[Cas9] mice at postnatal (P) age of 10 days (P10). For the control (Ctrl) group, the same dose of AAV9 virus containing pAAV9-sgRNA vector backbone without any sgRNA insertion was injected. The mice were then sacrificed for SC isolation and analysis after eight weeks (Fig. S8a). Successful DNA editing efficiency was confirmed for all four TFs (Supplementary Fig. 8b–e); consistently, knock down (KD) of each protein to various degrees (52.4%–83.6%) was detected by WB (Supplementary Fig. 7b–e) or IF (Supplementary Fig. 8f) despite unaltered mRNA levels (Supplementary Fig. 8g).

To examine the effect of the above AP-1 TF KD on SC activation, FISCs from each mutant were cultured for in vitro EdU assay (Supplementary Fig. 8a). Similar to *Atf3* KO, the *Fos*-KD cells showed accelerated SC activation while the *JunB*-KD displayed blunted activation; the *Atf4* or *FosB*-KD, on the other hand, did not have any impact (Supplementary Fig. 8h). When measuring Pax7+ cells on the ~2 month-old uninjured muscles, interestingly, reduced numbers were observed upon the *Atf4*, *Fos* or *JunB*-KD (Supplementary Fig. 8i), suggesting a loss of SC pool. Since the *Atf4* or *JunB*-KD did not lead to precocious SC activation, the SC number loss may be caused by other reasons considering the deletion occurred very early in the postnatal stage[64]. The *FosB*-KD did not have any impact on the Pax7+ SC pool.

To further investigate the impact on muscle regenerative ability, BaCl₂ was injected 7 weeks after the AAV injection (Supplementary Fig. 8a) and the regeneration was assessed by H&E (Supplementary Fig. 8j), eMyHC (Supplementary Fig. 8k), Pax7 (Supplementary Fig. 8l) and MyoD (Supplementary Fig. 8m) staining at 5 dpi. An impaired regeneration was revealed in the *Atf4* and *Fos*-KD presumably because of the exhaustion of SCs after its long-term deletion; The regeneration in the *JunB*-KD was also compromised probably due to a reduced SC pool; consistently, no impact on the regeneration was observed in the *FosB*-KD. Altogether, our results that these AP-1 family members may play diverse functions in SC activation and muscle regeneration.

## Discussion

In this study, we have elucidated the functional role of ATF3 in preventing premature activation of SCs in the mouse skeletal muscle. ATF3 and other AP-1 family members exhibit rapid and transient induction in SCs upon isolation induced early activation. Interestingly, the inducible *Atf3* deletion in SCs causes their precocious activation

and thus accelerates acute-injury induced regeneration, but the long-term *Atf3* depletion leads to the exhaustion of the SC pool in homeostatic muscle thus impairing its regeneration. The loss of ATF3 also provokes SC rapid activation during voluntary exercise and further enhances SC activation during endurance exercise. Mechanistically, ATF3 binds and directly regulates the transcription of Histone2B (H2B); the *Atf3* ablation leads to H2B downregulation, which accelerates nucleosome displacement during transcription to upregulate genes for SC activation. The H2B loss also results in genome instability and enhances replicative senescence in SCs. Lastly, a functional screening uncovers that several other AP-1 family TFs may also play diversified functions in SC activation and acute damage induced muscle regeneration. Therefore, this study has revealed a previously unknown role of ATF3 in SCs for preventing premature activation, thereby maintaining the SC pool and regenerative ability (Fig. 9i).

The competence of SCs to drive robust tissue regeneration is based on their ability to exit from their steady-state quiescent stage and pass to an activated state following stimuli encountered notably in traumatic or pathological conditions. Recent technical advances have permitted the exploration of both quiescence and activation and revealed a high level of complexity in the molecular regulation of these stages[12–14]. It is widely accepted that during mechanical enzymatic tissue dissociation, SCs undergo massive changes in transcription and histone modifications in fact enter an early activating stage; one of the signature events occurring in the early response to the disruption of their niche is the prominent induction of AP-1 members[12–14] but no mechanistic investigation was conducted so far. Our study is the first to provide a functional and mechanistic study of ATF3 induction in SCs. Our expression profiling recapitulates the early responsive nature of ATF3/AP-1: its rapid (within hours) and transient induction in SCs upon exposure to external stimuli including chemical/mechanical dissociation coupled with FACS-based purification, isolation of intact single muscle fibers, exercise-induced micro-trauma, and chemically induced muscle damage in vivo. The rapid induction of ATF3 upon stress induced by disrupting SC niche maintenance seemingly indicates a promoting role of ATF3 in the transition of SC quiescence to activation, it was thus surprising to observe a complete opposite phenotype upon deletion of *Atf3*. ATF3 loss clearly renders SCs to break quiescence and enter an early stage of activation even in the undisrupted homeostatic muscle. These cells can rapidly proceed to full activation once receiving stimuli from the growth medium in culture or acute injury in vivo. In acute damage induced muscle regeneration, this ability endows iKO SCs with enhanced regenerative potential; even with three rounds of injuries, iKO SCs still managed to achieve full regeneration of the damaged muscle. The precocious activation of iKO SCs was also confirmed in exercise settings. SCs have been implicated in exercise induced muscle growth but their exact contribution and mechanisms remain under explored due to many variables in exercise settings. Despite the voluntary exercise regimen did not cause SC activation in Ctrl mice, iKO SCs displayed evident activation and contribution to muscle hypertrophy growth in this type of exercise. In the more strenuous endurance exercise, iKO SCs

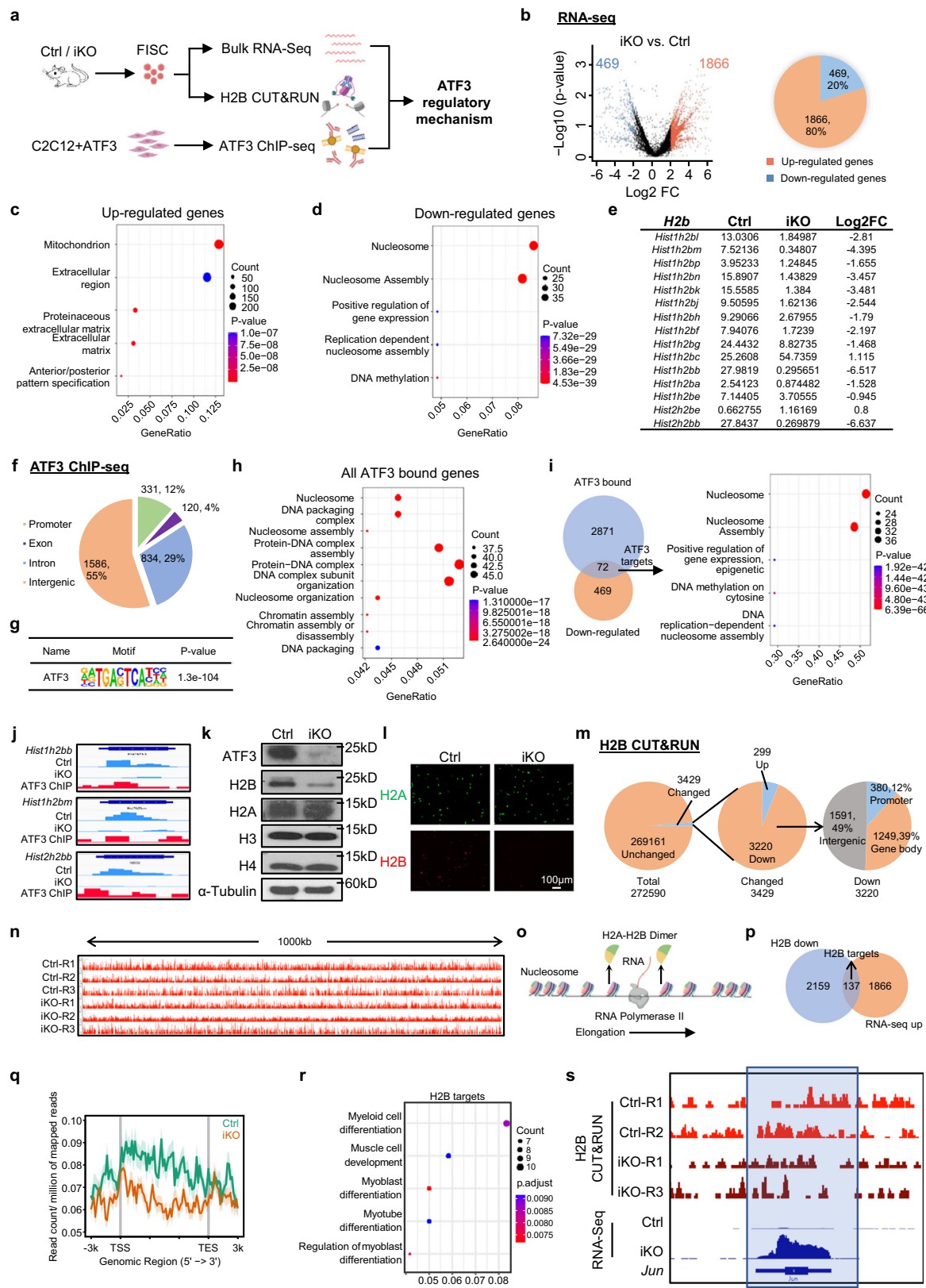

expectedly showed enhanced activating ability than Ctrl cells. It is not surprising that these iKO cells could easily undergo pseudo-regeneration in which they enter full activation, differentiation and fuse with fibers to deposit their nuclei centrally even without acute injury, therefore, depletion of SC pool occurred with long term deficiency of ATF3, rendering the regeneration impaired. Altogether, our

findings demonstrate the early activating status of iKO SCs thus exalted activating ability under various stimuli in multiple settings, therefore, the rapid induction of ATF3 in SCs by various stress signals appears to function to actively suppress the cells from proceeding to full activation. This therefore sets ATF3 distinct from previously characterized quiescence-maintaining factors that are usually highly

**Fig. 7 | ATF3 regulates *H2B* gene expression and nucleosome patterning.** **a** Schematic of the experimental design. Created with BioRender.com. **b** DEGs were identified from the above Ctrl vs. iKO RNA-seq. **c**, **d** GO analysis of the above up and down-regulated genes. **e** FPKM and Log2FC of *H2b* genes in iKO vs. Ctrl. **f** Genomic distribution of 2871 ATF3 binding peaks. **g** Enrichment of canonical ATF3 motifs in the above binding regions. **h** GO analysis for all the above genes with ATF3 binding. **i** Left: Venn diagrams showing the overlapping (72 genes) between the above identified ChIP-Seq target (2871) and the down-regulated genes (469). Right: GO analysis of the above 72 genes revealed an extreme enrichment of histone-related terms. **j** Genomic snapshots of 3 of the above identified *H2b* genes with ATF3 binding in their TSSs (ChIP-Seq tracks) and down-regulated by ATF3 deletion (RNA-Seq tracks). **k** ATF3, H2A, H2B, H3 and H4 proteins were measured by western blot in FISC from Ctrl and iKO mice. α-Tubulin was used as a loading control. **l** IF staining of H2A (green) and H2B (red) on the above FISCs. Scale bar: 100 μm; *n* = 3 mice per group. **m** Pie charts showing the number of bins with H2B changed (3429, 299 up and 3220 down) and unchanged (269161) (left and middle), and the genomic distribution of down-regulated H2B signals (right). **n** Genomic snapshots of a 1000 kb region Integrative Genomics Viewer (IGV) track of H2B signal in Ctrl vs. iKO on Chr4. **o** Schematic of the RNA Polymerase II elongation process during DNA transcription. Created with BioRender.com. **p** Venn diagrams showing the overlapping (137 genes) between the above regions with down-regulated H2B CUT&RUN signals (2159) and the up-regulated genes (1866) from the above (A) RNA-Seq. **q** Metaplots showing average H2B CUT&RUN signals 3 kb up- to down-stream of TSS. **r** GO analysis of the above 252 genes from (P). **s** Genomic snapshots of *Jun* gene showing down-regulated H2B CUT&RUN signal and up-regulated RNA-Seq signal in iKO vs. Ctrl. Regions with H2B signal reduction in iKO vs. Ctrl are highlighted in blue frame. Source data are provided as a Source Data file.

expressed in QSCs to actively preserve the quiescent state; these factors include PTEN[65,66], Notch[67–70], FoxO[11] or Rac[71] signaling, etc. Its function is also in contrast to those rapidly induced to facilitate the transition from quiescence to early activation such as Fos[40], mTORC1[10], or PI3K signaling[39]. In the homeostatic muscle, we believe ATF3 induction by minor stress such as daily movement, stretching, exercise-induced-micro-trauma, etc. serves to actively suppress activating program therefore indirectly preserving the deep quiescence of SCs. If exposed to more severe myofiber destruction such as BaCl$_2$ induced acute injury, the cells will need to down-regulate ATF3 and proceed with full activation; a small subset of cells probably preserves the expression of ATF3 and stemness and contribute to self-renewal in the later phase of regeneration. Consistently, we observed defect in self-renewal of iKO SCs after each round of three injuries. Of note, coincident with our theory, a recent report[72] shows that ATF3 plays an important role in maintaining hematopoietic stem cells (HSCs) self-renewal and preventing stress-induced exhaustion of HSCs despite no mechanistic insights being provided. Similarly, ATF3 deficiency leads to enhanced proliferation of HSCs under short-term stress but causes HSC exhaustion after long-term stress exposure. Therefore, it is possible that ATF3 plays previously unappreciated roles in stem cell homeostasis and regeneration in general.

To fathom the underlying mechanism of how ATF3 executes the above described functions in SCs, integrating RNA-Seq and ChIP-Seq, our findings uncovered interesting binding and regulation of ATF3 on Histone 2 transcription. Compared to the wealth of transcriptional regulatory mechanism of histone gene expression in budding yeast, much needs to be learned in mammals[58,59,73,74]. Here our findings reveal the direct transcriptional regulation of both clusters encoding *H2b* genes by ATF3. The binding of ATF3 on the divergent promoters of *H2b* genes on both chromosome 13 and 3 enables simultaneous regulation of a histone gene pair. Proper histone gene expression and histone protein synthesis are key to nucleosome assembly and composition which in turn governs chromatin structure and gene transcription. Rapid induction of ATF3 under stress may thus function to ensure H2B protein production and proper nucleosome organization for controlling the expression of genes to prevent cell activation. Loss of ATF3 expectedly caused an obvious loss of H2B protein in SCs, which interestingly did not result in genome-wide decrease of H2B enrichment by CUT&RUN, suggesting genome-wide nucleosome occupancy may not be largely impacted. It is likely the reduced amount of H2B may lead to formation of non-canonical nucleosomes[75], for example, so called half-nucleosomes consisting of one copy of each of the four core histones or hexasomes with two copies of H3/H4 and one copy of H2A/H2B. These possibilities can be investigated in detail in the future by additional approaches such as MNase digestion, etc. The function of these sub-nucleosomes in transcription is still unclear, it is possible that they prevalently exist in the iKO cells and alter the overall nucleosome structure and chromatin properties, therefore explaining the overall transcriptional activation occurring in the cell. In addition,

the existence of these sub-nucleosomes may also affect genomic stability and increase the propensity for DNA damage thus cellular senescence, which was indeed observed in the iKO cells (Fig. 7). Aging-coupled histone loss also results in elevated levels of DNA strand breaks and genomic instability in budding yeast[73,74], it will thus be interesting to examine whether ATF3 loss will accelerate SC aging which is known to contribute to muscle tissue aging[76–78].

In addition to ATF3, several other AP-1 family members showed rapid induction in FISCs, suggesting their potential functionality in the transition from quiescence to early activation. Indeed, the in vivo functional screening revealed rather diversified roles of ATF4, FOS, FOSB and JUNB in SC activation and muscle regeneration. While ATF4 and FOS appear to play similar functions as ATF3 in preventing precocious activation, JUNB may have an opposite action. FOSB loss, on the other hand, did not have a detectable effect on SCs in the screening. Indeed, Wang et al.[39] demonstrates that c-Jun is a key transcriptional target of the PI3K/mTORC1 signaling axis essential for SC quiescence exit upon muscle injury. Taglietti et al.[38] also shows that JUNB is an activator of fetal myogenesis. Nonetheless, it is interesting to point out that a recent study[40] demonstrates that rapid and transient induction of FOS is required to direct SC early activation events needed for efficient regeneration. FOS is found to be heterogeneously induced in a subset of SCs which exhibit a greater propensity for entering the early stages of activation. Loss of FOS in SCs thus impairs their capacity to enter the cell cycle. We suspect the discrepancy from our study could arise from the different systems used to delete FOS. Our in vivo editing system has been proven to be effective in the initial screening for key regulators of SC activities but more definitive investigation can be performed using an inducible genetic mouse model in the future. It is also possible that ATF3 is mainly induced in the subset of SCs with high propensity for deep quiescence and self-renewal while FOS is expressed in a subset of SCs with a greater propensity for entering the early stage of activation. Loss of ATF3 in the subset of SCs provokes them to become pro-activating and regenerative. This reinforces the increasingly recognized molecular and functional heterogeneity within the SC population which equips the cells with adaptive potentials in response to different demands[10].

## Methods

### Animal studies
All animal handling procedures, protocols and experiments ethics approval was granted by the CUHK AEEC (Animal Experimentation Ethics Committee) under the Ref No. 16-166-MIS and 21-254-MIS. The mice were maintained in animal room with 12 h light/12 h dark cycles, temperature (22–24 °C), and humidity (40–60%) at animal facility in CUHK. For all animal-based experiments, at least three pairs of littermates or age-matched mice were used.

The Tg: *Pax7-nGFP* mouse strains[37], *Pax7CreER* (*Pax7tm1(cre/ERT2)Gaka*)[79]; *ROSAEYFP* reporter mice and *Pax7Cre* (*Pax7tm1(cre)Mrc*)[48]; *ROSAEYFP* reporter

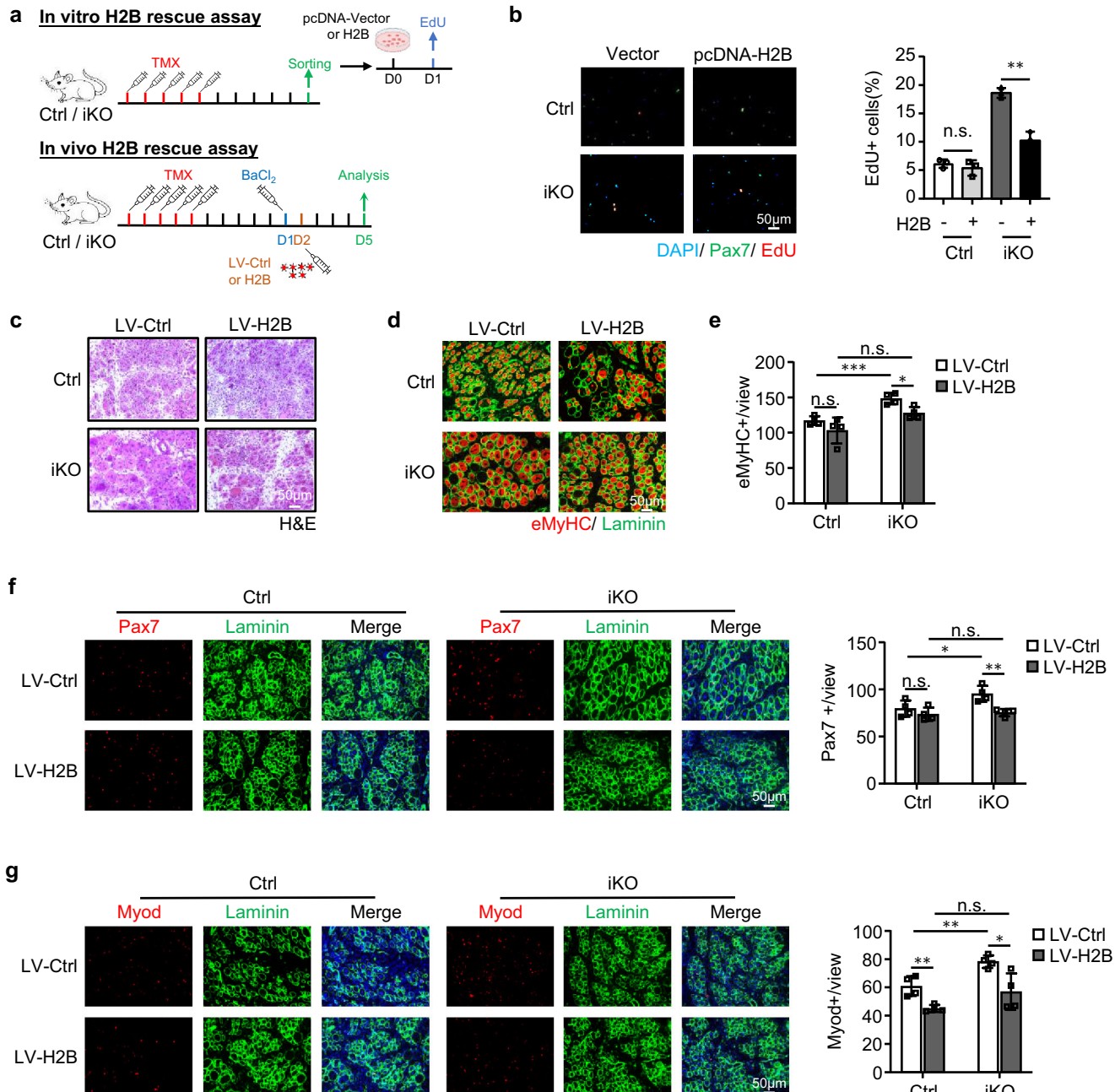

**Fig. 8 | H2B mediates ATF3 function in SC activation and muscle regeneration.**
**a** Upper: Schematic for overexpressing H2B in vitro. FISCs from Ctrl or iKO mice were transfected with a control (Ctrl) or pCDNA-H2B plasmid and EdU assay was performed for assessing SC activation. Lower: Schematic for overexpressing H2B in vivo via lenti virus. H2B expressing lentivirus was injected into Ctrl or iKO TA muscles 1 day after BaCl₂ injury and the muscles were collected 5 days after infection for analysis. **b** Left: The above transfected cells were cultured for 24 h before treated with EdU for 6 h; EdU positive cells were stained and quantified. Scale bar: 50 μm; *n* = 3 mice per group. *p* = 0.44 and 0.0013. **c** H&E staining of the above TA muscles collected at 5 dpi after infection. Scale bar: 50 μm. *n* = 4 mice per group. **d** IF staining of eMyHC (red) and Laminin (green) was performed on the above TA muscles. Scale bar: 50 μm. **e** The numbers of eMyHC+ fibers per view were

quantified. *n* = 4 mice per group. From left to right, *p* = 0.20, 0.00081, 0.052 and 0.012. **f** Left: IF staining of Pax7 (red) and Laminin (green) was performed on the above TA muscles. Scale bar: 50 μm. Right: the numbers of Pax7+ cells per view were quantified. *n* = 4 mice per group. From left to right, *p* = 0.33, 0.041, 0.67 and 0.0052. **g** Left: IF staining of Myod (red) and Laminin (green) was performed on the above TA muscles. Scale bar: 50 μm. Right: the numbers of Myod+ cells per view were quantified. *n* = 4 mice per group. From left to right, *p* = 0.0053, 0.0058, 0.12 and 0.020. All the bar graphs are presented as mean ± SD. Student's *t* test (two-tailed unpaired) was used to calculate the statistical significance (**b**, **e–g**): *\*p* < 0.05, *\*\*p* < 0.01, *\*\*\*p* < 0.001. n.s. no significance. Source data are provided as a Source Data file.

mice were kindly provided by Dr. Zhenguo WU (Hong Kong University of Science and Technology). The *Atf3 ^{fl/fl}* mouse strain was kindly provided by Prof. Tsonwin HAI (Ohio State University, USA)[46]. The C57BL wildtype mice were purchased from LASEC (Laboratory Animal Services Centre) of CUHK. The *Atf3* inducible conditional KO mice (*Atf3* iKO) with EYFP reporter (Ctrl: *Pax7^{CreER/+}; ROSA^{EYFP/+}; Atf3+/+*, iKO: *Pax7^{CreER/+}; ROSA^{EYFP/+}; Atf3^{fl/fl}*) were generated by crossing *Pax7^{CreER}; ROSA^{EYFP}* with *Atf3^{fl/fl}* mice. The *Atf3* conditional KO mice (*Atf3* cKO) with EYFP reporter (Ctrl: *Pax7^{Cre/+}; ROSA^{EYFP/+}; Atf3+/+*, cKO: *Pax7^{Cre/+}; ROSA^{EYFP/+}; Atf3^{fl/fl}*) were generated by crossing *Pax7^{Cre}; ROSA^{EYFP}* with

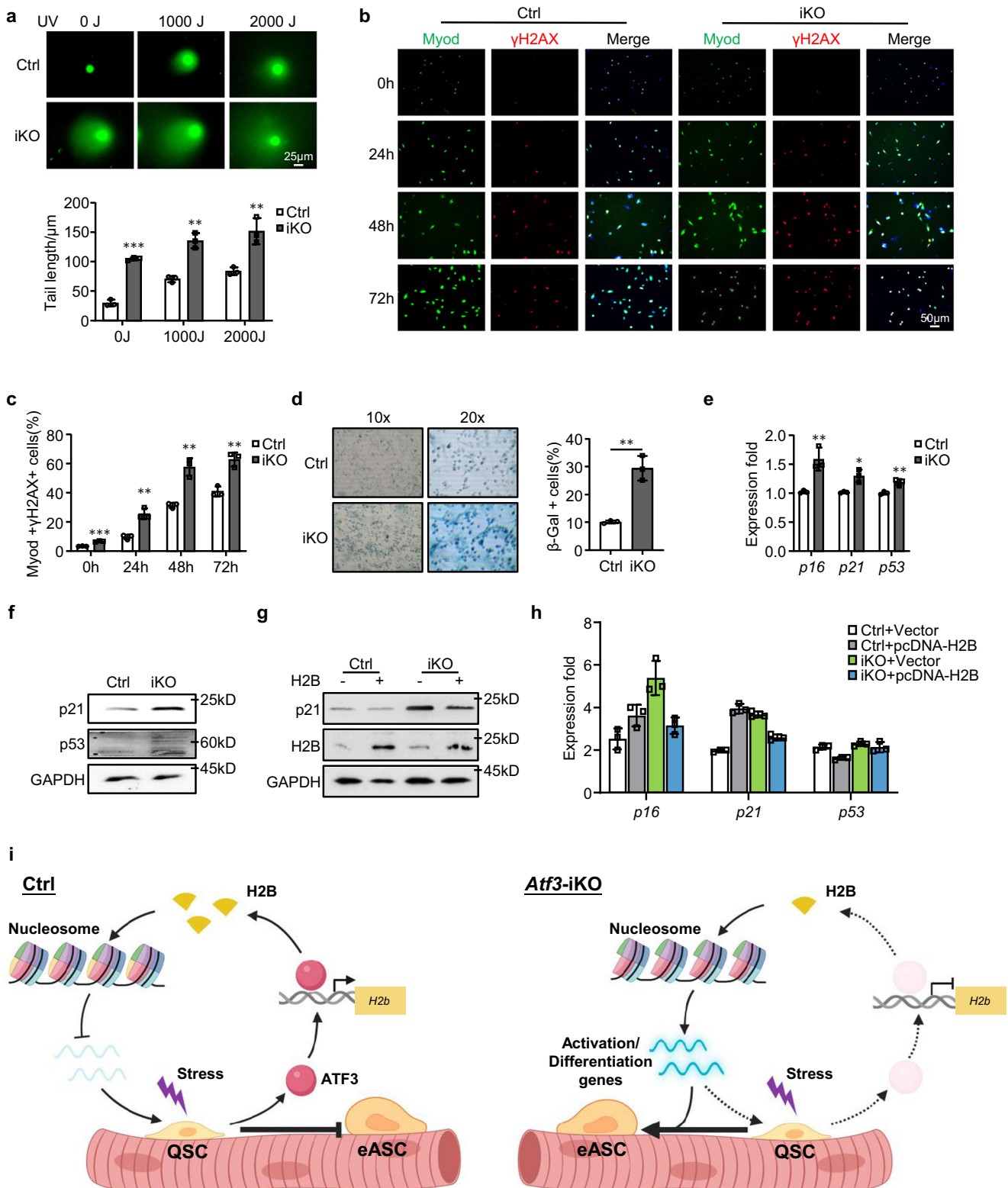

*Atf3fl/fl* mice. To induce Cre-mediated *Atf3* deletion, Tamoxifen (TMX) (T5648, Sigma) was injected intraperitoneally at 2 mg per 20 g body weight for consecutive 5 days. The *Pax7Cre* dependent *Rosa26Cas9-EGFP* knock-in mice were generated as described[64]. Primers used for geno-typing are shown in Supplementary Data 5.

To induce acute muscle injury, 50 μl of 1.2% BaCl₂ (dissolved in sterile demineralized water) was injected into TA muscle of ~2-month-old mice. Muscles were harvested at designated time points for subsequent histological or biochemical studies. For in vivo EdU incorporation assay, 10 mg EdU diluted in 100 ul PBS was injected intraperitoneally, TA muscles and myofibers were harvested 12 h later, or 0.6 mg EdU diluted in 100 ul PBS were IP injected for consecutive 5 days; TA muscles and myofibers were harvested 21 days later, followed by 4% PFA fixation. EdU-labeled cells were visualized using click chemistry with an Alexa Fluor® 594 conjugated azide. Images were captured with a fluorescence microscope (Leica).

**Fig. 9 | H2B loss increases DNA damage and senescence in *Atf3* iKO SCs. a** Comet assay was performed on Ctrl or iKO SCs after 0, 1000 or 2000J UV treatment. Scale bar: 25 µm. *n* = 3 mice per group. *p* = 0.000029, 0.0012, and 0.0071. **b** IF staining of γH2AX (red) and Myod (green) was performed on SCs from Ctrl or iKO mice after cultured for 0, 24, 48 or 72 h. Scale bar: 50 µm. **c** Quantification of the percentages of Myod+/γH2AX+ cells; *n* = 3 mice per group. *p* = 0.00014, 0.0024, 0.0021 and 0.0031. **d** β-Gal (blue) staining was performed on SCs from Ctrl or iKO mice after cultured for 9 days. *n* = 3 mice per group. *p* = 0.0017. **e** Expression of the selected senescence marker genes including *p16*, *p21* and *p53* in SCs from Ctrl and iKO was quantified by RT–qPCR. *p* = 0.0078, 0.011 and 0.0061. **f** p21 and p53 protein levels were detected by Western blotting in SCs from Ctrl and iKO mice. GAPDH was used as a loading control. **g, h** SCs from Ctrl and iKO mice were transfected with an H2B-

overexpressing or vector control plasmid. 96 h after transfection, expression of the indicated genes was detected by Western blotting and RT–qPCR. GAPDH was used as a loading control. **i** Schematic model depicting the functional role of ATF3 in preventing SC precocious activation. In homeostatic muscle, the rapid induction of ATF3 upon minor stress promotes *H2b* expression to maintain proper nucleosome positioning and suppress the expression of activation genes, thus preventing precocious activation of SCs. Upon ATF3 loss, H2B deficiency occurs which leads to altered nucleosome positioning and up-regulation of activation genes, causing SCs to break quiescence and enter early activating stage. Created with BioRender.com. All the bar graphs are presented as mean ± SD. Student's *t* test (two-tailed unpaired) was used to calculate the statistical significance (**a**, **c**–**e**): *\*p* < 0.05, *\*\*p* < 0.01, *\*\*\*p* < 0.001. n.s. no significance. Source data are provided as a Source Data file.

---

For the voluntary wheel running exercise, ~2-month-old mice were housed individually for 4 weeks in polycarbonate cages with 12-cm-diameter wheels equipped with optical rotation sensors (Yuyan Instrument, ARW) After 5 doses of TMX injection, the mice were firstly trained by the above settings for 7 days and followed by consecutive 21 days of exercise. For endurance exercise, ~2-month-old mice were adapted to a treadmill (Panlab, Harvard Apparatus, 76-0895) with a 5° incline at a speed of 20 cm/s for 60 min. After 5 doses of TMX injection, the mice were firstly trained by the above settings for 5 days and followed by consecutive 10 days of exercise. For non-exercised control, mice were housed individually in standard polyethylene cages without access to any exercise wheel or treadmill.

### CRISPR/Cas9 mediated in vivo genome editing

The in vivo genome editing via CRISPR/Cas9 was performed according to previously described instructions[64]. Briefly, Cre-dependent *Pax7*$^{Cas9}$ mouse was generated through crossing *Pax7*$^{Cre}$ mouse[48] with homozygous *Rosa*$^{Cas9-eGFP}$ mouse[80], resulting in the labeling of all Pax7 derived cells (muscle lineage) with eGFP. In vitro validated sgRNA pairs targeting each AP-1 family TF were generated from a U6-driven AAV9 backbone using muscle-tropic AAV9 as the delivery vector. The *Pax7*$^{Cas9}$ mice were injected intramuscularly with a single dose ($5 \times 10^{11}$ vg per mouse) of AAV9-sgRNA at postnatal day 10 (P10) and were analyzed 8 weeks later.

### Satellite cell isolation and culture

Satellite cells were sorted based on established methods[81,82]. Briefly, entire hindlimb muscles from mice were digested with collagenase II (LS004177, Worthington, 1000 units per 1 ml) for 90 min at 37 °C, the digested muscles were then washed in washing medium (Ham's F-10 medium (N6635, Sigma) containing 10% horse serum, heat-inactivated (HIHS, 26050088, Gibco, 1% P/S) before SCs were liberated by treating with Collagenase II (100 units per 1 ml) and Dispase (17105-041, Gibco, 1.1 unit per 1 ml) for 30 min. The suspensions were passed through a 20G needle to release myofiber-associated SCs. Mononuclear cells were filtered with a 40-µm cell strainer and sorted by BD FACSAria IV (fluorescence-activated cell sorting) with the selection of the positive GFP fluorescence signal. BD FACSVerse flow cytometer, BD FACSAria Fusion Cell Sorter and BD FACSDiva (Version 8.0.1, BD Biosciences) were used for the acquisition of flow cytometry data. Coverslips and cultural wells were coated with poly-D-lysine solution (p0899, Sigma) at 37 °C for overnight and then coated with extracellular matrix (ECM) (E-1270, Sigma) at 4 °C for at least 6 h. FACS-isolated SCs were seeded in coated wells and cultured in Ham's F10 medium with 10% HIHS, 5 ng ml$^{-1}$ β-FGF (PHG0026, Thermo Fisher Scientific) and 1% P/S.

### Single myofiber isolation and culture

As described before[83,84], single myofibers were isolated from the EDL (extensor digitorum longus) muscles of adult mice. The EDL was dissected and digested in the digestion solution (800U Collagenase type II in 1 ml DMEM medium) at 37 °C for 75 min with a shake of 70 rpm.

After digestion, the single myofibers were released by gentle trituration with Ham's F-10 medium containing 10% HIHS and 1% P/S and cultured in this medium for designated time points.

### Cell line culture

C2C12 mouse myoblast cell line (CRL-1772) and 293T cells (CRL-3216) were obtained from ATCC (American Type Culture Collection) and cultured in DMEM medium (Gibco, 12800-017) with 10% FBS (fetal bovine serum, Gibco, 10270-106) and 1% P/S (Penicillin-Streptomycin, 10,000 U/ml, Gibco,15140-122) at 37 °C in cell incubator with 5% $CO_2$. The C2C12 cells were induced to differentiate when they reached an 80–90% confluence, by altering the GM to DM (differentiation medium) which consisted of DMEM medium, 2% HS (Horse Serum, Gibco, 16050114) and 1% P/S under the same culture environment. Lentivirus particles expressing sgRNSs of AP-1 family members were packaged in 293T cells as previously described[64].

### Cell proliferation assay

EdU incorporation assay was performed following the instruction of Click-iT® Plus EdU Alexa Fluor® 594 Imaging Kit (C10639, Thermo Fisher Scientific). Growing cells on coverslips were incubated with 10 µM EdU for a designated time before the fixation with 4% PFA for 20 min. EdU-labeled cells were visualized using "click" chemistry with an Alexa Fluor® 594-conjugated azide and cell nuclei were stained by DAPI (Life Technologies, P36931). Images were captured with a fluorescence microscope (Leica).

### SA-β-galactosidase staining

Cellular senescence was evaluated by β-galactosidase activity using β-galactosidase Senescence Kit (Cell signaling, #9860) after being cultured for designated times. Briefly, satellite cells were fixed for 10–15 min followed by washing in PBS. The cells were then incubated with β-galactosidase staining solution at 37 °C at least overnight in a dry incubator without $CO_2$. The cells were then observed under a microscope for the development of blue color which represents the existence of β-galactosidase, a significant feature of senescent cells.

### Comet assay

The alkaline comet assay was conducted to measure the DNA damage level in satellite cells. Sorted satellite cells were treated with UV radiation at different dosages and the comet assay was operated immediately after irradiation through a Comet Assay Kit (Abcam, ab238544) according to the manufacturer's instructions. The amount of DNA damage was indicated by the tail moment which is a measure of DNA fragments with a Leica fluorescence microscope.

### Plasmids

To construct ATF3 expression plasmid, a 545 bp DNA fragment of *Atf3* CDS (coding DNA sequence) region was amplified and cloned into a pcDNA3.1+ vector between KpnI and XbnI. To construct H2B expression plasmid, a 493 bp DNA fragment of mouse *hist2h2bb* gene full-

length was amplified and cloned into a pcDNA3.1+ vector between HindIII and KpnI. To generate *Atf4*, *Fos*, *FosB* and *JunB* in vivo knockdown mice, two sgRNAs for each target gene were designed by CRISPOR[85] and cloned into PX458 vector at the BbsI site[64]. All primers used for plasmid construction and lentivirus production are presented in Supplementary Data 5.

## Lentivirus packaging and infection

To construct lentiviral vectors for overexpressing ATF3 or H2B, full-length of ATF3 and H2B were separately cloned into the pRLenti vector (Addgene, USA) according to the manufacturer's instructions. 8.5 µg psPAX2 (Addgene, USA), 4.07 µg PDM2.G (Addgene, USA) and 7.6 µg pRLenti-GFP/pRLenti-ATF3/pRLenti-H2B together with 40 µl Lipo3000 were transfected into 293T cells cultured in a T75 flask. The cells were incubated for 6 h before the media was replaced with 10 ml of DMEM complete. Virus particles were harvested at 24 and 48 h post transfection, centrifuged at -500 × *g* for 5 min and filtered through a 0.45 µm PES filter, then stored at −80 °C for use. For infection of C2C12 cells, 500 µl of the above generated virus were added to 1.5 ml of culture medium together with polybrene at the final concentration of 10 µg/ml. For infection of mouse muscle, 100 µl virus particles were injected into TA muscles 1 day after BaCl$_2$ injury. All primers used for plasmid construction and lentivirus production are presented in Supplementary Data 5.

## RNA extraction and qRT-PCR

Total RNAs from cells or tissues were extracted using TRIzol reagent (Invitrogen, 15596018) according to the manufacturer's instructions. cDNAs were synthesized under the manufacturer's instructions of HiScript® II Reverse Transcriptase Kit with gDNA wiper (Vazyme, R223-01). Real-time PCR was performed to quantify the expression level of mRNAs by Luna® Universal qPCR Master Mix (NEB, M3003E) and LightCycler ®480 Real-Time PCR System (Roche). GAPDH (glyceraldehydes 3-phosphate dehydrogenase) or 18s RNA were used as internal controls for normalization. The primers used were listed in Supplementary Data 5. The relative fold changes compared to control groups were calculated by the classical ΔΔCt method.

## Immunoblotting, immunofluorescence, and immunohistochemistry

For Western blot, cells or tissues were harvested, washed with PBS, and lysed in RIPA buffer supplemented with protease inhibitor cocktail, PIC (88266, Thermo Fisher Scientific) for 30 min on ice. The protein concentration was determined using a Bradford protein assay kit (Bio-Rad). Whole-cell lysates were subjected to SDS−PAGE, and protein expression was visualized using an enhanced chemiluminescence detection system (GE Healthcare, Little Chalfont, UK) as described before[86]. The following dilutions were used for each antibody: ATF3 (Santa Cruz Biotechnology, c-188x; 1:5000), H2A (Abcam, ab177308; 1:2500), H2B (Abcam, ab1790; 1:2500), H3 (Santa Cruz Biotechnology, sc-8654; 1:4000), H4 (Abcam, ab177840; 1:2000), α-tubulin (Santa Cruz Biotechnology, sc-23948; 1:5000) and GAPDH (Sigma-Aldrich, G9545-100UL;1:5000).

For immunofluorescence staining, cultured cells or myofibers were fixed in 4% PFA for 15 min and permeabilized with 0.5% NP-40 for 10 min. Then cells were blocked in 3% BSA for 1 h followed by incubating with primary antibodies overnight at 4 °C and secondary antibodies for 1 h at RT. Finally, the cells or myofibers were mounted with DAPI to stain the cell nucleus and images were captured by a Leica fluorescence microscope. Primary antibodies and dilutions were used as follows PAX7 (Developmental Studies Hybridoma Bank, PAX7-S-1ML; 1:50), MyoD (Dako, M3512; 1:1000. Santa Cruz Biotechnology, sc-304x; 1:2000), MyoG (Santa Cruz Biotechnology, sc-12732; 1:200), ATF3 (Santa Cruz Biotechnology, c-188x; 1:2000), H2A (Abcam,

ab177308; 1:1000), H2B (Abcam, ab1790; 1:1000), γ-H2AX (Biolegend, 613401; 1:200), ATF4 (Santa Cruz Biotechnology, 390063; 1:200), FOS (Santa Cruz Biotechnology, sc-8047; 1:200), FOSB (Santa Cruz Biotechnology, 398595; 1:200) and JUNB (Santa Cruz Biotechnology, 8051; 1:200).

For immunohistochemistry, in brief, slides were fixed with 4% PFA for 15 min at room temperature and permeabilized in ice-cold menthol for 6 min at −20 °C. Heat-mediated antigen retrieval with a 0.01 M citric acid (pH 6.0) was performed for 5 min in a microwave. After 4% BBBSA (4% IgG-free BSA in PBS; Jackson, 001-000-162) blocking, the sections were further blocked with unconjugated AffiniPure Fab Fragment (1:100 in PBS; Jackson, 115-007-003) for 30 min. The biotin-conjugated anti-mouse IgG (1:500 in 4% BBBSA, Jackson, 115-065-205) and Cy3-Streptavidin (1:1250 in 4% BBBSA, Jackson, 016-160-084) were used as secondary antibodies. Primary antibodies and dilutions were used as follows PAX7 (Developmental Studies Hybridoma Bank, PAX7-S-1ML; 1:50), MyoD (Dako, M3512; 1:500), eMyHC (Developmental Studies Hybridoma Bank, F1.652; 1:200) and Laminin (Sigma, L9393-100UL; 1:800). All fluorescent images were captured with a fluorescence microscope (Leica).

H&E (Hematoxylin-eosin) staining on TA muscle sections was performed according to a protocol described before[87]. Section slides were first stained in hematoxylin for 10 min followed by rinsing thoroughly under running tap water for at least 3 min. Then section slides were immersed in 0.2% acid alcohol for 1 s and immediately rinsed under running tap water. Next, section slides were stained in eosin for 2 min followed by rinsing and dehydrating in graded ethanol and Xylene. Finally, slides were mounted by DPX and observed under a normal microscope.

## RNA-seq and data analysis

For RNA-seq, RNAs from satellite cells were extracted by Trizol reagent and delivered to Beijing Genome Institute (BGI), Hong Kong with dry ice. Data released by BGI from sequenced fragments were mapped to mouse genome mm9[88] using TopHat. The abundance levels of transcripts were defined by cufflinks[89] in the form of FPKM (reads per kilobase per million reads mapped). Differentially expressed genes (DEGs) between samples were identified by the change of expression levels using a threshold of log2FoldChange > 2.

## ChIP-seq and data analysis

Briefly, cells were cross-linked in 1% formaldehyde and processed according to previously described[90,91]. Ten µg of antibodies against ATF3 (Santa Cruz Biotechnology, c-188x), or normal mouse IgG (Santa Cruz Biotechnology, sc-2025) was used for immunoprecipitation. Immunoprecipitated genomic DNA was resuspended in 20 µl of water. For DNA library construction, a NEBNext® Ultra™ II DNA Library Prep Kit for Illumina® (NEB, E7645S) was used according to the manufacturer's instructions. Bioanalyzer analysis and qPCR were used to measure the quality of DNA libraries including the DNA size and purity. Finally, DNA libraries were sequenced on the Illumina Genome Analyzer II platform.

The raw data were first pre-processed by initial quality assessment, adapters trimming, and low-quality filtering and then mapped to the mouse reference genome (mm9) using bowtie2[92], and only the non-redundant reads were kept. The protein DNA-binding peaks (sites) were identified using MACS2[93] with input (IgG) sample as the background. During the peak calling, candidate peaks were compared with the background, dynamic programming was used to determine λ of Poisson distribution, and the *P*-value cutoff was set to 0.001 for ATF3 ChIP-Seq experiment.

## CUT&RUN and date analysis

CUT&RUN assay was conducted using 250,000 satellite cells with the CUT&RUN assay kit (Cell Signaling Technology, 86652). In brief, FISCs

were harvested and washed by cell wash buffer, then bound to con-canavalin A-coated magnetic beads. Digitonin Wash Buffer was used for permeabilization. After that, cells were incubated with 5 µg of H2B antibody (Abcam, ab1790) overnight at 4 °C with shaking. Then, cell-bead slurry was washed with Digitonin Wash Buffer and incubated with Protein A-MNase for 1 h at 4 °C with shaking. After washing with Digitonin Wash Buffer, CaCl2 was added into the cell-bead slurry to initiate Protein A-MNase digestion, which was then incubated at 4 °C for half an hour. Then 2x Stop Buffer was added to the reaction to stop the digestion. CUT&RUN fragments were released by incubation for 30 min at 37 °C followed by centrifugation. After centrifugation, the supernatant was recovered, and DNA purification was performed by using Phenol/Chloroform (Thermo). For DNA library construction, a NEBNext® Ultra™ II DNA Library Prep Kit for Illumina® (NEB, E7645S) was used according to the manufacturer's instructions. Bioanalyzer analysis and qPCR were used to measure the quality of DNA libraries including the DNA size and purity. Finally, DNA libraries were sequenced on the Illumina Genome Analyzer II platform.

The raw reads were first pre-processed by quality assessment, adapters trimming, and low-quality filtering, and then were aligned to the mouse reference genome (mm9) using Bowtie2[92], and only-redundant reads were kept. For the analysis of genome-wide differential H2B enrichment, we calculated the mean signal for H2B within each bin with 10 kb using the function "multiBigwigSummary bins" in deeptools[94]. Bins with average signal lower than 1 were removed. A threshold of 1.5-fold change was used to classify changed or unchanged bins between the ATF3-iKO samples and Ctrl samples. Only bins shared in at least 2 out of 3 replicates were considered as high-confidence bins. To analyze the differential H2B enrichment, we compared the signals on the promoter and gene body between ATF3-iKO and Ctrl samples. For each promoter or gene body, we divided the region evenly into 40 bins, and calculate the mean signals of each bin. The $p$ value cutoff was set to 0.05 for the two-side of the Mann–Whitney U test of comparing signals of ATF3-iKO and Ctrl. $p$ value < 0.05 and fold change >1.5 were used to define significantly different. with the Mann–Whitney U test. $p$ value < 0.05 and fold change >1.5 were used to define significantly different. Only genes (including promoter or gene body) found in at least 2 out of 3 replicates were considered as high-confidence.

### Gene Ontology analysis
ClusterProfiler[95] was used for the Gene Ontology (GO) analysis with Entrez gene IDs converted from DAVID[96] tool as inputs. The adjusted $p$ or $p$ values were reported with the GO terms. The $p$ value was calculated by the hypergeometric distribution. Benjamini–Hochberg method was used to adjust the $p$ value. And $q$ value was set to 0.05.

### Statistical and reproducibility
Data were analyzed using Excel (version 2208 Build 16.0.15601.20676; Microsoft 365MSO, Redmond, WA) and GraphPad Prism (version 8; GraphPad Software, San Diego, CA). Data were represented as the average of at least three biologically independent samples ± SD. The statistical significance was assessed by the Student's two-tailed unpaired $t$ test by Excel software. Significance was described as n.s, not significant; *$p$ < 0.05; **$p$ < 0.01 and ***$p$ < 0.001. Representative images of at least three independent experiments are shown in Figs. 1f–h and 7k and Supplementary Figs. 2b–d, 3a, c and 6d–g. Representative images of two independent experiments are shown in Fig. 9f and Supplementary Fig. 7h, k.

### Reporting summary
Further information on research design is available in the Nature Portfolio Reporting Summary linked to this article.

## Data availability
RNA-Seq, ATF3 ChIP-Seq and H2B CUT&RUN data generated in this study have been deposited in Gene Expression Omnibus (GEO) data-base under the accession codes GSE205170, GSE205314, GSE205324 and GSE205548. The data supporting the findings of this study are available from the corresponding author on reasonable request. Source data are provided with this paper.

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

## Acknowledgements

This work was supported by National Key R&D Program of China to H.W. (project code: 2022YFA0806003); General Research Fund (GRF) from Research Grants Council (RGC) of the HongKong (HK) Special Administrative Region, China to H.W. (project codes: 14100620, 14105823, 14106521 and 14115319 to H.W.; 14105123, 14103522, 14120420 and 14120619 to H.S.); Theme-based Research Scheme (TRS) from RGC (project code:T13-602/21-N); Collaborative Research Fund (CRF) from RGC (project code: C6018-19GF); Health and Medical Research Fund (HMRF) from Health Bureau of HK to H.W. (project codes: 10210906 and 08190626); the National Natural Science Foundation of China (NSFC) to H.W. (project codes: 82172436 and 31871304); the research funds from Health@InnoHK program launched by Innovation Technology Commission, the Government of HK to H.W.; CUHK Strategic Seed Funding for Collaborative Research Scheme (SSFCRS) to H.W.; Area of Excellence Scheme (AoE) from RGC (project code: AoE/M-402/20).

## Author contributions

S.Z. designed and performed most experiments; Y.H., F.Y. and Y.D. analyzed RNA-Seq, ATF3 ChIP-Seq and H2B CUT&RUN data; L.H., Y.L. and C.E.W. provided technical supports; K.M.C. supervised CUT&RUN assay; H.S. supervised computational analyses; S.Z., Y.H., F.Y., T.X. and H.W. wrote the manuscript, with inputs from all authors.

## Competing interests

The authors declare no competing interests.
