## [Peer Review File · Nature Communications]

REVIEWER COMMENTS

Reviewer #1 (Remarks to the Author):

The manuscript brings highly interesting new insight into the control of muscle stem cells quiescence and senescence, with extensive experimental data. The finding that ATF3 controls H2B expression thus preventing genome instability and replicative senescence

in SCs is unexpected, novel and important.

Has requested, I did especially look at the cut & run data. The signal to noise ratio appears to be the problem. The image in Fig 6n has low resolution, but the image in Fig 6s on the Eln gene doesn't really show peaks consistent with nucleosomes. Just a lot of noise across the gene. This makes me wonder about the quality of the data set. That being said, the cumulative data shows there is a decrease at the 5' end of the gene around the TSS (Fig 6q). This is the one thing I would say is true on the Eln gene in Figure 6S.

So, overall, I would recommend that ideally the experiment should be repeated to get better signal to noise ratio before being able to state there is a loss of H2B at specific areas of the genome.

Reviewer #2 (Remarks to the Author):

Skeletal muscles have the remarkable capacity to undergo regenerative growth in response to injury or other external stimuli. Muscle regeneration relies on activation and expansion of skeletal muscle stem cells, also known as satellite cells (SCs), residing beneath the basal lamina of myofibers, followed by differentiation into multinucleated myotubes. The mechanism for preserving the SCs population has become an attractive avenue of research for the discovery of possible therapeutic interventions for many muscle degeneration disorders. The manuscript by Zhang et al. provides some interesting aspects for SCs area. The study put multiple sets of data in mice and cells together into a story. However, the reviewer found numerous issues in the logic, data quality and conceptual novelty.

First, I have a hard time reckoning with the main conclusion "ATF3 preserves skeletal muscle stem cell quiescence". What the authors showed in this paper was that ATF3 was not expressed/or expressed at very low levels in SCs at quiescence state (QSC), and ATF3 is only rapidly induced upon SCs activation, even though the authors showed some data suggesting that deletion of ATF3 in SCs accelerates acute injury-induced regeneration. The authors also showed many difference phenotypes between the "Short-term" and "long-term" ATF3 iKO, this could be due to many reasons, either experimental systems (such as Tamoxifen system) or chronic secondary compensation. Moreover, the authors claimed that ATF3 is induced upon SCs activation and ATF3 prevents SCs activation. However, no data to demonstrate that ATF3 overexpression is sufficient to inhibit SCs activation. In this reviewer's view, no data support the concept in this paper.

The second issue is the link to H2B. If the authors claim the linear link between ATF3 and H2B, the critical experiment would be to demonstrate that ATF3 overexpression sufficiently prevents SCs activation, and genetic loss of H2B blocks the process.

The third issue is about data figures: there are many data provided; however there are several shortcomings in the presentation of the data, data interpretation, and importantly, the lack of high quality data.

Indeed, the conceptual advance that this manuscript will bring forward is also being questioned.

Major Points:

1. As discussed above, the authors concluded that ATF3 preserves muscle stem cell quiescence. However, in this reviewer's view, the data did not support this concept in this paper. What the authors showed in this paper was that ATF3 was not expressed/or expressed at very low levels in SCs at quiescence state (QSC), and ATF3 is only rapidly induced upon SCs activation, even though the authors show some data suggesting that deletion of ATF3 in SCs accelerates acute injury-induced regeneration. The authors also showed many difference phenotypes between the "Short-term" and "long-term" ATF3 iKO, this could be due to many reasons, either experimental systems (such as Tamoxifen system) or chronic secondary compensation.
2. The authors claimed that ATF3 is induced upon SCs activation, and ATF3 prevents SCs activation. However, no data to demonstrate that ATF3 overexpression is sufficient to inhibit SCs activation.
3. Although the author did some interesting experiment to suggest that H2B loss in SCs results in increased genome instability and cellular senescence. In this reviewer's view, if the authors claim the linear link between ATF3 and H2B, the critical experiment would be to demonstrate that ATF3 overexpression sufficiently prevents SCs activation, and genetic loss of H2B blocks the process.
4. There are several shortcomings in the presentation of the data and data interpretation, and importantly, the lack of high quality data. For example, all IF staining data (multiple replicates of the experiment) need quantification to show statistical significance; very poor quality images in Fig. 2e, Fig. 2g, Fig. 2n and Fig. 3g....., and all the poor quality laminin stainings; the Reviewer is confusing, In Fig.1f, why the Pax7 staining is so low in QSC compare to others in the same panel? In Fig. 4d, the image suggests increased Pax7 levels in iKO at 5dpi, but the quantification data show decreased in iKO at 5dpi.
5. Furthermore, N = 3 throughout the paper, with such a small sample size for mice studies, how can the authors be certain their data is reproducible?
6. The study follows work previously published by the same group, using the same dataset, the authors describe that ATF3 was induced in FISC compared to QSCs, but indeed this is also known and reported previously by other group (Machado L, et al., Cell Rep. 2017;21(7):1982-1993.). Moreover, previously study has also reported that ATF3 regulates stem cells self-renewal (Liu Y, et al., Front Cell Dev Biol. 2020;8:585771), a paper that the authors have cited. Thus, the conceptual novelty is marginal in this journal.

7. By comparing the pre-fixed and non-fixed SCs (indeed quite an artificial system), the authors claimed that ATF3 is induced upon SCs activation. But as the authors said, AP-1 family members are very sensitive to a variety of cellular stresses and quickly induced by even slight disturbance. Therefore, it would be very important to examine whether ATF3 is induced in SCs during injury induced muscle regeneration in vivo.

8. The authors used the term "resistance exercise" to study the SCs activation. However, in page 12, the authors described a typical endurance exercise protocol that uses a treadmill set at a 5° incline and a speed of 20 cm/sec. This protocol is not resistance exercise. The reviewer is very confusing, since endurance exercise do not trigger SC activation and muscle hypertrophy. It would be much better to study the SCs activation of these mice with a protocol of real resistance exercise (Cui D, et al., FASEB J. 2020;34(6):7330-7344.).

9. SCs undergo a self-renewal process and returns to the quiescent stage to replenish the SC pool. The authors found that deletion of ATF3 in SCs increased muscle weight and fiber size during 1st, 2nd and 3rd injury. They also showed that Pax7+ cells were decreased in ATF3 KO after 3rd injury. If the authors claim that ATF3 KO exhausts SC pool, it would be at very least to examine the SC pool (Pax7+ cells) after 1st injury and 2nd injury as well.

10. The similar engraftment assay (Fig. 2l,m) need to be done to examine the regenerative ability of the iKO SCs in the "long-term" iKO mouse models.

11. Figure S7. The authors performed some functional screening using the Pax7 Cas9/AAV-sgRNA mediated in vivo genome editing system. Although there are many data provided, the data do not support authors claim due to the low editing efficiency and the none-inducible SC specific deletion.

Minor:

1. The authors mislabel The Fig.1e, 1d in the text. page 6: "this was also confirmed by RT-qPCR (Fig.1e)", Fig.1e not RT-PCR .

2. Many errors in the text. e.g. page 6: "which was also n confirmed by RT-qPCR (Fig. S1a)."

Reviewer #3 (Remarks to the Author):

This manuscript describes a novel role of the activating transcriptional factor 3 (ATF3) in muscle stem cells called satellite cells. The authors first used RNA-seq to show and immunostaining to confirm that Atf3 expression is transiently elevated in activated satellite cells (Fig. 1). They then generated conditional KO mice to delete Atf3 gene in quiescent satellite cells using the Pax7-CreER driver. They show that the cKO exhibited accelerated regeneration initially but regeneration declines upon repeated injury at the 3rd injury, associated with reduced myogenic cells. These results suggest a defect in self-renewal that sequentially reduces satellite cell number and regenerative capacity. Figure 3 use single fiber culture and in vivo regeneration model to confirm there is a defect in self-renewal of Atf4 KO cells as manifested by reduced Pax7+/Myod- cells, even though the cell proliferation is elevated. They then

show in figure 4 a regenerative effect in response to a single bout of injury 6 months after Atf3 deletion. In Figure 5 they show that Atf3 KO satellite cells activate more readily in response to exercise. In figure 6, they use RNA-seq, CUT&RUN and ChIP-seq to bring about the idea that loss of ATF3 suppress transcription of the H2b clusters. Figure 7 demonstrate that the Atf3 null satellite cells display aggravated stress-induced DNA damage and accelerated aging phenotypes. Together, the study uses a number of omics, conditional KO, in vivo injury and exercise models to demonstrate for the first time a key role of ATF3 in satellite cells and muscle regeneration. The results are striking and based on large amount of solid data.

I only have a main concern, which is related to the interpretation of the results that can be fixed by rewording. The title (ATF3 Preserves Skeletal Muscle Stem Cell Quiescence by Preventing Precocious Activation), the abstract, results all indicate that ATF plays a role in quiescent cells but the data appear to suggest a role of ATF3 in self-renewal (i.e. it is elevated in activated cells to force the cell back to a quiescent state). I think this concept must be clarified throughout the manuscript.

Related to the previous point on ATF3 expression: Fig 1g showing expression of ATF3 in satellite cells on fresh isolated myofiber suggesting its expression in quiescent cells (even though fiber isolation takes 1-2 hours that could activate cells, it takes more than two hours for a gene to be transcribed and then made into proteins). Therefore, my worry is that in situ fixation may induce artifacts in gene expression. It would have been nice if sections of non-injured and injured muscles can be used to validate or reject the conclusion that ATF3 expression is only induced by activation.

Still related to data interpretation: If as the authors concluded, ATF3 is not expressed in quiescent satellite cells (Fig 1b, c, d, f), then why would knock out of a non-expressed gene affect satellite cells over time? This does not make sense. It only makes sense if ATF3 is indeed expressed in the quiescent satellite cells. The authors argue that there are various stress conditions that induce activation of satellite cells but what is the threshold of such stress? Why would normal use (movements) does not induce the (activation) stress whereas exercise induce stress? Again, if ATF3 function to push stress activated satellite cells back to a quiescent state, then its function is more in line with the self-renewal function in HSC. You cannot declare that a protein function to enforce quiescence while it is not expressed in the quiescent satellite cells.

Minor:

Subtitles of the results section do not effectively capture the results.

In Figure 5I, H2B is completely lost in the iKO. It is intriguing that the phenotype is so "mild" given the complete loss of H2B. On another note, the histone proteins are thought to be very stable, even a

complete shutdown of H2b transcription should not deplete all H2B proteins without extensive cell division. Please clarify.

Abstract: “Here we report that AP-1 family member ATF3 preserves the SC quiescence by preventing their premature activation. Atf3 is rapidly and transiently induced in SCs upon activation.” Without any context the two sentences appear to contradict. If ATF3 preserves quiescence then induction of Atf3 should enforce quiescence but not lead to activation. I suggest that these should be rewritten.

Similarly, the ending sentence of the first paragraph in Results: “Taken together, our findings show that ATF3 and several other AP-1 family members are rapidly and transiently induced during early SC activation, suggesting the potential roles of ATF3 and other family members in the regulation of SC quiescence or early activation.” Without reading discussion this sentence does not make sense at all.

As satellite cells were increased by 80% in the Pax7-Cre KO in embryonic myoblasts, is the muscle hypertrophic in the young mice?

Also in p12: “Altogether the above results suggest an essential role of ATF3 induction in preventing precocious SC activation,..” So far the study has never shown that ATF3 induction prevents precocious activation.

P13: “These results demonstrate that the ATF3 loss decreases H2b gene expression”. This is counter-intuitive as Atf3 KO leads to activation and proliferation, which require generation of new histone protein.

Full name of ATF3 should be provided at the first occurrence. ATF3 is known to activate gene expression and it is puzzling why most genes are upregulated in Atf3 KO satellite cells. Some discussion might be helpful.

Abstract: “which reduction accelerates nucleosome displacement and gene transcription required for SC activation”. This sentence appears to have grammar issue. Change to “whose reduction”

The Atf3 KO phenotype is very similar to, but milder than, Pten KO phenotypes (PMID: 27880908, PMID: 28094257). In discussion, the authors discussed other quiescent regulators and PTEN should be included. Also, there are reports that ATF3 and PTEN may phenocopy and regulate each other (PMID: 25531328, PMID: 27308526). These should also be discussed.

Point by point responses

Reviewer #1:

The manuscript brings highly interesting new insight into the control of muscle stem cells quiescence and senescence, with extensive experimental data. The finding that ATF3 controls H2B expression thus preventing genome instability and replicative senescence in SCs is unexpected, novel and important.

1.1. Has requested, I did especially look at the cut & run data. The signal to noise ratio appears to be the problem. The image in Fig 6n has low resolution, but the image in Fig 6s on the Eln gene doesn't really show peaks consistent with nucleosomes. Just a lot of noise across the gene. This makes me wonder about the quality of the data set. That being said, the cumulative data shows there is a decrease at the 5' end of the gene around the TSS (Fig 6q). This is the one thing I would say is true on the Eln gene in Figure 6S. So, overall, I would recommend that ideally the experiment should be repeated to get better signal to noise ratio before being able to state there is a loss of H2B at specific areas of the genome.

A: We thank the reviewer for the constructive comment. We agree that the CUT&RUN experiment should be repeated to improve the signal to noise ratio. We have now repeated the experiment in FISCs from Ctrl and Atf3 iKO mice and provided data from three replicates. This time, we set a more stringent cutoff to remove low bin signals to improve the signal to noise ratio. The findings from analyzing the new datasets remain largely the same as our original one. The average H2B CUT&RUN signals were largely unaltered in the iKO vs. Ctrl. but 93.9% of the altered bins showed decreased H2B enrichment (Fig. m). When intersecting with the original RNA-Seq data, this time 137 up-regulated genes showed decreased H2B signals on their promoters or gene bodies and are known to be associated with SC activation and differentiation (Fig. r). These newly added results can be found on page 14-15 of the revised text.

Reviewer #2:

Skeletal muscles have the remarkable capacity to undergo regenerative growth in response to injury or other external stimuli. Muscle regeneration relies on activation and expansion of skeletal muscle stem cells, also known as satellite cells (SCs), residing beneath the basal lamina of myofibers, followed by differentiation into multinucleated myotubes. The mechanism for preserving the SCs population has become an attractive avenue of research for the discovery of possible therapeutic interventions for many muscle degeneration disorders. The manuscript by Zhang et al. provides some interesting aspects for SCs area. The study put multiple sets of data in mice and cells together into a story. However, the reviewer found numerous issues in the logic, data quality and conceptual novelty.

2.1 *As discuss above, the authors concluded that ATF3 preserves muscle stem cell quiescence. However, in this reviewer's view, the data did not support this concept in this paper. What the authors showed in this paper was that ATF3 was not expressed/or expressed at very low levels in SCs at quiescence state (QSC), and ATF3 is only rapidly induced upon SCs activation, even though the authors show some data suggesting that deletion of ATF3 in SCs accelerates acute injury-induced regeneration. The authors also showed many difference phenotypes between the “Short-term” and “long-term” ATF3 iKO, this could be due to many reasons, either experimental systems (such as Tamoxifen system) or chronic secondary compensation.*

A: Thanks for the critical comment. The reviewer 3 raised a similar question in his/her comment 3.1 regarding the use of “preserves quiescence” to describe ATF3 function in SCs. We now agree that it may not be accurate to describe its function as quiescence maintenance. We agree that actively maintaining the quiescence can only be executed by factors that are expressed in quiescent stage such as PTEN, Notch, FoxO and Rac. In the case of ATF3, its direct function is to suppress early activation and indirectly preserves SC quiescence. We have now revised throughout the text accordingly.

Additionally, we don't think the observed phenotypes from our ATF3 iKO mice arise from experimental errors. As shown in Fig. 2c and Fig S4e, ATF3 protein is completely depleted in both short-term and long-term settings.

2.2 *The authors claimed that ATF3 is induced upon SCs activation, and ATF3 prevents SCs activation. However, no data to demonstrate that ATF3 overexpression is sufficient to inhibit SCs activation.*

A: Thanks for the constructive comment. We agree that is it critical to show gain of function evidence to provide solid evidence to support our argument that ATF3 actively suppresses SC activation. As suggested, we have now performed the ATF3 overexpression both in vitro and in vivo. In vitro we over-expressed ATF3 by a lenti-virus in FISC. Both EdU assay and Pax7/Myod double staining were then performed to demonstrate that indeed SC activation was inhibited by AFT3 over-expression (Fig. 3j-k). In vivo, we over-expressed ATF3 by intramuscular injection of the lenti-virus at 1 dpi in both Ctrl and iKO mice and found the overexpression obviously delayed regeneration in Ctrl mice and also blunted the accelerated muscle regeneration in iKO mice. (Fig. 3l-q and Fig. S3c-e). The above findings thus strengthened our claim that ATF3 prevents SC activation. The newly added results can be found on page 10 of the revised text.

2.3 *Although the author did some interesting experiment to suggest that H2B loss in SCs results in increased genome instability and cellular senescence. In this reviewer's view, if the authors claim the liner link between ATF3 and H2B, the critical experiment would be to demonstrate that ATF3 overexpression sufficiently prevents SCs activation, and genetic loss of H2B blocks the process.*

A: Thanks for the great suggestion. We agree that it is necessary to demonstrate the functional link between ATF3 and H2B by rescue experiment. As described above in 2.2, we have now performed ATF3 over-expression to demonstrate it can prevent SC activation. We also made attempts to knockdown H2B in vitro or in vivo only to realize this is not possible because of the large quantity of H2B produced in cells. As shown in Fig. S6c, histone encoding genes are typically organized into multigene clusters and H2B protein is encoded by 2 gene clusters with 15 on Chr13 forming a Hist1h2b cluster and 2 on Chr3 forming a Hist2h2b cluster. It is thus impossible to decrease histone levels by siRNA knockdown¹. In fact, it is also rare to find studies using siRNAs to knockdown H2B. To circumvent the difficulty, we instead over-expressed H2B in vitro and in vivo. In vitro we found that over-expressing H2B by a transfecting a pcDNA-H2B plasmid in FISCs indeed repressed accelerated activation of Atf3 iKO cells but no impact on Ctrl cells (Fig. 7a-b). And in vivo, we also over-expressed H2B by intramuscular injection of H2B expressing lentivirus and found that H2B overexpression indeed blunted the accelerated regeneration in Atf3-iKO mice (Fig. 7c-g). Altogether these findings validate that H2B loss indeed mediates the precocious SC activation and enhanced muscle regeneration. The newly added results can be found on page 15 of the revised text.

2.4 *There are several short-comings in the presentation of the data and data interpretation, and importantly, the lack of high quality data. For example, all IF staining data (multiple replicates of the experiment) need quantification to show statistical significance; very poor quality images in Fig. 2e, Fig. 2g, Fig. 2n and Fig. 3g....., and all the poor quality laminin stainings; the Reviewer is confusing, In Fig. 1f, why the Pax7 staining is so low in QSC compare to others in the same panel? In Fig. 4d, the image suggests increased Pax7 levels in iKO at 5dpi, but the quantification data show decreased in iKO at 5dpi.*

A: Thanks for the critical comment. We apologize for the poor quality of some IF images in our original submission. We have now provided high quality images in Fig. 2e, Fig. 4c, Fig. 4f. To better show the laminin staining, we have now provided separate images in Fig. S2i, Fig. S2j, Fig. S3a, Fig. S4a, Fig. S4b, Fig. S5h, Fig. S5i, Fig. S5j and Fig. S5k. We apologize for the low Pax7 staining on QSC in Fig. 1f and have now repeated the experiment to provide images with high quality. In the original Fig.4d we apologize for using the non-representative images and have now replaced it to be consistent with the quantification data in Fig. S4a.

2.5 *Furthermore, N = 3 throughout the paper, with such a small sample size for mice studies, how can the authors be certain their data is reproducible?*

A: Thanks for the critical comment. We agree that N=3 is a small number for animal experiments. Still we argue that we performed multiple sets of experiments to support each conclusion. For example, to demonstrate the defect of Atf3 iKO in SC activation, we not only used in vitro cultured SCs but also single myofibers as well as in vivo muscle sections to conduct the EdU assay; our conclusions were therefore drawn from more than 3 mice. Nevertheless, to solidify our conclusions, following the suggestion, we have now increased the sample size to five mice for experiments

described in Fig. 2d-f, Fig. 2k, Fig. 2m, Fig. 2n, Fig. 3a-h, Fig. 4b-g, Fig. 5b-g, Fig. 5i-n, Fig. S2c-h, Fig. S3g and Fig. S5a-b. With the newly added mice, all our original conclusions remain unaltered. The corresponding figure legends have also been changed accordingly.

2.6 The study follows work previously published by the same group, using the same dataset, the authors describe that ATF3 was induced in FISC compared to QSCs, but indeed this is also known and reported previously by other group (Machado L, et al., Cell Rep. 2017;21(7):1982-1993.). Moreover, previously study has also reported that ATF3 regulates stem cells self-renewal (Liu Y, et al., Front Cell Dev Biol. 2020;8:585771), a paper that the authors have cited. Thus, the conceptual novelty is marginal in this journal.

A: Thanks for the critical comment but we have to disagree with the reviewer's opinion. Yes, we acknowledge that the induction of ATF3 in FISC compared to QSC has been noticed and reported by several groups (Almada AE, et al., Nat Rev Mol Cell Biol. 2016;17(5):267–79; Machado L, et al., Cell Rep. 2017;21(7):1982-1993; van Velthoven CTJ, et al., Cell Rep. 2017;21(7):1994–2004; van den Brink SC, et al., Nat Methods. 2017;14(10):935–6; Almada AE, et al., Cell Rep. 2021;34(4):108656; Barutcu, A.R., et al., Skelet Muscle. 2022;12(1):20.)²⁻⁷. However, none of these reports investigated the function of ATF3 induction in detail. We are the first one to dive in and provide a functional and mechanistic investigation of ATF3 in SCs. We also acknowledge that a prior study has reported that ATF3 regulates stem cells self-renewal (Liu Y, et al., Front Cell Dev Biol. 2020;8:585771), however, this study was performed in HSCs and the findings are different from ours. For example, the study shows that ATF3 is down-regulated after stress stimulation while in our case it is induced in FISC. ATF3 deficiency leads to enhanced proliferation and expansion of long-term repopulating hematopoietic stem cells (LT-HSCs) upon short-term chemotherapy or irradiation. The long-term reconstitution capability of LT-HSCs is dramatically impaired after a series of bone marrow transplantations, indicating that ATF3 plays a protective role in stress hematopoiesis to maintain HSC self-renewal⁸. These are completely different aspects of stem cell activities compared to our study. Moreover, this study provides no mechanistic insights into how ATF3 functions in HSCs while ours thoroughly elucidates the underlying mechanism of how ATF3 regulates H2B expression to function in SC activation. Therefore, we argue that our findings provide sufficient conceptual novelty not only for the field of skeletal muscle regeneration but also the stem cell field in general considering this is the first study showing an induced protein can function to actively prevent not promote stem cell activation. The above points have been included in the revised text on page 18 and page 20.

2.7 By comparing the pre-fixed and non-fixed SCs (indeed quite an artificial system), the authors claimed that ATF3 is induced upon SCs activation. But as the authors said, AP-1 family members are very sensitive to a variety of cellular stresses and quickly induced by even slight disturbance. Therefore, it would be very important to examine whether ATF3 is induced in SCs during injury induced muscle regeneration in vivo.

A: Thanks for the great suggestion. We agree it is important to show ATF3 induction in vivo. We have now performed IF staining on both uninjured and injured TA muscle (at various time after injury) sections. Expectedly, our data (Fig. 1h) demonstrated that ATF3 is not expressed in uninjured muscles. At 1dpi we observed increased ATF3 expression and some staining did not merge with Pax7+ cell, which is in line with its ubiquitous expression in multiple cells⁹ induced by injury¹⁰. At 2dpi ATF3 expression was highly increased from Pax7+ cells, concomitant with the activation stage of the SCs. These results thus provide solid evidence to support ATF3 is induced upon SC activation during muscle regeneration. The newly added results can be found on page 6-7 of the revised manuscript.

2.8 *The authors used the term "resistance exercise" to study the SCs activation. However, in page 12, the authors described a typical endurance exercise protocol that uses a treadmill set at a 5° incline and a speed of 20 cm/sec. This protocol is not resistance exercise. The reviewer is very confusing, since endurance exercise do not trigger SC activation and muscle hypertrophy. It would be much better to study the SCs activation of these mice with a protocol of real resistance exercise (Cui D, et al., FASEB J. 2020;34(6):7330-7344.).*

A: Thanks for the critical comment. We agree that the described exercise protocol should be called "endurance exercise". We are very sorry for the mistake and have now made changes in the manuscript. However, we would like to point out that endurance exercise does trigger SC activation and muscle hypertrophy according to previous reports. For example, Cisterna B. et. al showed that, endurance training of mice on a treadmill increases the activation of SCs as well as their capability to differentiate into myotubes¹¹. Fry CS. et.al; Macaluso F. et.al. Joannis S. et.al and Mackey AL et.al all demonstrated that after different types of aerobic exercises, human muscle also showed increased SC activation, myonuclear content and fiber diameter¹²⁻¹⁵. Therefore, we think it is reasonable to use the endurance exercise in our study to demonstrate the enhanced activating ability of ATF3-iKO cells in an exercise setting. Switching to a resistance training model may take extra time which may delay the publication of the study. We hope the reviewer will find this acceptable.

2.9 *SCs undergo a self-renewal process and returns to the quiescent stage to replenish the SC pool. The authors found that deletion of ATF3 in SCs increased muscle weight and fiber size during 1st, 2nd and 3rd injury. They also showed that Pax7+ cells were decreased in ATF3 KO after 3rd injury. If the authors claim that ATF3 KO exhausts SC pool, it would be at very least to examine the SC pool (Pax7+ cells) after 1st injury and 2nd injury as well.*

A: Thanks for the great suggestion. We have now examined Pax7+ cell numbers after the 1st, 2nd and 3rd round of injury and found that SC numbers were indeed decreased after each round of injury (Fig.2k), supporting our conclusion that ATF3 deletion leads to the exhaustion of SC pool during injury induced muscle regeneration. The newly added results can be found on page 10 of the revised text.

2.10 *The similar engraftment assay (Fig. 2l,m) need to be done to examine the regenerative ability of the iKO SCs in the “long-term” iKO mouse models.*

A: Thanks for the suggestion. We have now performed the engraftment assay after the long term ATF3 deletion. As illustrated in Fig. 4h, 4 months after TMX injection, YFP+ donor SCs were collected from the Ctrl or iKO mice and injected into the recipient nude mice which were pre-injured 1 day before the engraftment. 21 days later, the injected TA muscles were harvested for IF staining. A higher number of YFP+ myofibers were observed in the mice transplanted with the iKO vs. Ctrl SCs (Fig. 4i), suggesting that enhanced regenerative ability of the iKO cells persists after long term ATF3 loss and the impaired muscle regeneration indeed arises from the reduced cell pool. The newly added results are included on page 11 of the revised text.

2.11 *Figure S7. The authors performed some functional screening using the Pax7 Cas9/AAV-sgRNA mediated in vivo genome editing system. Although there are many data provided, the data do not support authors claim due to the low editing efficiency and the none-inducible SC specific deletion.*

A: Thanks for the comment. The purpose of the functional screening in Fig. S7 is to provide initial evidence for the differential roles that AP-1 family members may play in muscle regeneration. The Pax7 Cas9/AAV-sgRNA mediated in vivo genome editing system permits efficient gene editing in quiescent SCs and was used in many of our prior publications¹⁶⁻²⁰, it is thus an acceptable approach for such screening. In Fig. S8 we demonstrate that the editing efficiency is in fact very good, 84% for ATF4, 52% for FOS, 67% for FOSB and 64% for JUNB. We acknowledge that this is not an inducible SC specific deletion which will provide more solid evidence for their roles in SCs. We have now revised the text on page 21 to point out this only serves as an initial screening and further investigation can be performed using an inducible genetic mouse model.

2.12 *The authors mislabel The Fig.1e, 1d in the text. page 6: “this was also confirmed by RT-qPCR (Fig.1e)”, Fig.1e not RT-PCR .*

A: Thanks for the comment and we apologize for the mistake. We have now corrected the labeling in Fig. 1e.

2.13 *Many errors in the text. e.g. page 6: “which was also n confirmed by RT-qPCR (Fig. S1a).”*

A: Thanks for the comment. We have now gone through the entire text and corrected the errors.

Reviewer #3:

This manuscript describes a novel role of the activating transcriptional factor 3 (ATF3) in muscle stem cells called satellite cells. The authors first used RNA-seq to show and immunostaining to

confirm that *Atf3* expression is transiently elevated in activated satellite cells (Fig. 1). They then generated conditional KO mice to delete *Atf3* gene in quiescent satellite cells using the *Pax7-CreER* driver. They show that the cKO exhibited accelerated regeneration initially but regeneration declines upon repeated injury at the 3rd injury, associated with reduced myogenic cells. These results suggest a defect in self-renewal that sequentially reduces satellite cell number and regenerative capacity. Figure 3 use single fiber culture and in vivo regeneration model to confirm there is a defect in self-renewal of *Atf4* KO cells as manifested by reduced *Pax7+*/*Myod-* cells, even though the cell proliferation is elevated. They then show in figure 4 a regenerative effect in response to a single bout of injury 6 months after *Atf3* deletion. In Figure 5 they show that *Atf3* KO satellite cells activate more readily in response to exercise. In figure 6, they use RNA-seq, CUT&RUN and ChIP-seq to bring about the idea that loss of ATF3 suppress transcription of the H2b clusters. Figure 7 demonstrate that the *Atf3* null satellite cells display aggravated stress-induced DNA damage and accelerated aging phenotypes. Together, the study uses a number of omics, conditional KO, in vivo injury and exercise models to demonstrate for the first time a key role of ATF3 in satellite cells and muscle regeneration. The results are striking and based on large amount of solid data.

3.1 I only have a main concern, which is related to the interpretation of the results that can be fixed by rewording. The title (*ATF3 Preserves Skeletal Muscle Stem Cell Quiescence by Preventing Precocious Activation*), the abstract, results all indicate that ATF plays a role in quiescent cells but the data appear to suggest a role of ATF3 in self-renewal (i.e. it is elevated in activated cells to force the cell back to a quiescent state). I think this concept must be clarified throughout the manuscript.

A: Thanks for the great comment. The Reviewer 2 raised a similar concern in his/her comment 2.1. We now agree that it may not be accurate to describe ATF3 function as quiescence maintenance. We agree that actively maintaining quiescence can only be executed by factors that are expressed in quiescent stage such as PTEN, Notch, FoxO and Rac. In the case of ATF3, its direct function is to suppress early activation and indirectly preserves SC quiescence as a consequence. We have not changed the title to “ATF3 Induction Prevents Precocious Activation of Skeletal Muscle Stem Cell by Regulating H2B Expression” and revised several places throughout the text accordingly. As for its role in SC self-renewal, we did observe reduced *Pax7+* cells after each of the three rounds of injuries which hinted a possible function in self-renewal upon acute regeneration. This is included on page 10 of the text.

3.2 Related to the previous point on ATF3 expression: Fig 1g showing expression of ATF3 in satellite cells on fresh isolated myofiber suggesting its expression in quiescent cells (even though fiber isolation takes 1-2 hours that could activate cells, it takes more than two hours for a gene to be transcribed and then made into proteins). Therefore, my worry is that in situ fixation may induce artifacts in gene expression. It would have been nice if sections of non-injured and injured muscles can be used to validate or reject the conclusion that ATF3 expression is only induced by activation.

A: Thanks for the great suggestion. This comment is also raised by reviewer 2 (comment 2.7). As stated above in our answer, we agree it is important to show ATF3 induction in vivo. We have now performed IF staining on both uninjured and injured TA muscle (at various time after injury) (Fig. 1h). Expectedly, our data indeed demonstrated that ATF3 is not expressed on uninjured muscle. At 1dpi we observed increased ATF3 expression and some staining did not merge with Pax7+ cell, which is in line with its ubiquitous expression in multiple cells⁹. At 2dpi ATF3 expression was highly increased from Pax7+ cells, concomitant with the activation stage of the SCs. These results thus provide solid evidence to support ATF3 is induced upon SC activation during muscle regeneration. The newly added results can be found on page 6-7 of the revised text.

3.3 *Still related to data interpretation: If as the authors concluded, ATF3 is not expressed in quiescent satellite cells (Fig 1b, c, d, f), then why would knock out of a non-expressed gene affect satellite cells over time? This does not make sense. It only makes sense if ATF3 is indeed expressed in the quiescent satellite cells. The authors argue that there are various stress conditions that induce activation of satellite cells but what is the threshold of such stress? Why would normal use (movements) does not induce the (activation) stress whereas exercise induce stress? Again, if ATF3 function to push stress activated satellite cells back to a quiescent state, then its function is more in line with the self-renewal function in HSC. You cannot declare that a protein function to enforce quiescence while it is not expressed in the quiescent satellite cells.*

A: Thanks for the great comment. As stated in the answer to your comment 3.1, we now agree that it may not be accurate to describe its function as quiescence maintenance. We agree that the direct function in actively maintaining the quiescence can only be executed by factors that are expressed in quiescent stage such as PTEN, Notch, FoxO and Rac. In the case of ATF3, its direct function is to suppress early activation and consequently preserves SC quiescence. We have not changed the title and revised several places throughout the text.

Secondly, it is a great question as what kind of stress would induce activation of SCs. There is an increasing body of literature on the association of exercise and SC activation. It is now clear that short-term, non-strenuous, voluntary exercise does not disturb SC quiescence but resistance training or more intense endurance exercise can induce SC activation. It is believed that endurance exercises and strength training cause muscle fiber micro-damage thus SC activation to repair damaged fiber²¹. In homeostasis, it is believed that SCs can also activate and proliferate sporadically to replace myofibres damaged by normal use of muscle e.g. daily movements or activities²². In our next chapter of the investigation, it will be interesting to further test how and what kind of stress signaling from daily movement or normal exercise triggers SC activation via ATF3 or other early response genes. We have now added the above points on page 19-20 of the revised text. We thank the reviewer for all these thought-provoking comments.

3.4 *Subtitles of the results section do not effectively capture the results.*

Thanks for the comment. We only made a slight change on the first subtitle on page 6 as we feel the original subtitles are OK to recapitulate the results. We are open to further suggestions from the reviewer.

3.5 *In Figure 5I, H2B is completely lost in the iKO. It is intriguing that the phenotype is so “mild” given the complete loss of H2B. On another note, the histone proteins are thought to be very stable, even a complete shutdown of H2b transcription should not deplete all H2B proteins without extensive cell division. Please clarify.*

A: Thanks for the questions. In fact, we have discussed this in the original submission. We believe proper histone gene expression and histone protein synthesis are key to nucleosome assembly and composition which in turn governs chromatin structure and gene transcription. Loss of ATF3 expectedly caused an obvious loss of H2B protein in SCs, which interestingly did not result in genome-wide decrease of H2B enrichment by CUT&RUN, suggesting genome-wide nucleosome occupancy may not be largely impacted. It is likely the reduced amount of H2B may lead to formation of non-canonical nucleosomes, for example, so called half-nucleosomes consisting of one copy of each of the four core histones or hexasomes with two copies of H3/H4 and one copy of H2A/H2B. The function of these sub-nucleosomes in transcription is still unclear, it is possible that they prevalently exist in the iKO cells and alter the overall nucleosome structure and chromatin properties, therefore explaining the overall transcriptional activation occurring in the cell. In addition, the existence of these sub-nucleosomes may also affect genomic stability and increase the propensity for DNA damage thus cellular senescence, which was indeed observed in the iKO cells. The above points can be found on page 20-21 of the revised text.

3.6 *Abstract: “Here we report that AP-1 family member ATF3 preserves the SC quiescence by preventing their premature activation. Atf3 is rapidly and transiently induced in SCs upon activation.” Without any context the two sentences appear to contradict. If ATF3 preserves quiescence, then induction of Atf3 should enforce quiescence but not lead to activation. I suggest that these should be rewritten.*

A: Thanks for the suggestion. As stated earlier, we agree it is not accurate to describe ATF3 function as quiescence maintenance, we have revised the abstract to better summarize our findings.

3.7 *Similarly, the ending sentence of the first paragraph in Results: “Taken together, our findings show that ATF3 and several other AP-1 family members are rapidly and transiently induced during early SC activation, suggesting the potential roles of ATF3 and other family members in the regulation of SC quiescence or early activation.” Without reading discussion this sentence does not make sense at all.*

A: Thanks for the comment, we have revised this sentence on page 7 of the revised text.

3.8 *As satellite cells were increased by 80% in the Pax7-Cre KO in embryonic myoblasts, is the muscle hypertrophic in the young mice?*

A: Thanks for the great question. We have now measured the TA muscle weight of the 1-month-old mice to show that there was a decrease in cKO compared with control (Fig. S4h). We have also quantified the fiber size but found no significant difference (Fig. S4i). It is thus hard to conclude that muscle hypertrophy occurs in the young cKO mice. The results are included on page 11 of the revised manuscript.

3.9 Also in p12: *“Altogether the above results suggest an essential role of ATF3 induction in preventing precocious SC activation,..” So far the study has never shown that ATF3 induction prevents precocious activation.*

A: Thanks for the critical comment. as shown in our answer to above comment 2.2, we agree that it is necessary to show overexpressing ATF3 can repress SC activation. As suggested, we have now performed the ATF3 overexpression both in vitro and in vivo. In vitro we over-expressed ATF3 by a lenti-virus in FISC. Both EdU assay and Pax7/Myod double staining were then performed to demonstrate that indeed SC activation was inhibited by AFT3 over-expression (Fig. 3j-k). In vivo, we over-expressed ATF3 by intramuscular injection of the lenti-virus at 1 dpi in both Ctrl and iKO mice and found the overexpression obviously delayed regeneration in Ctrl mice and also blunted the accelerated muscle regeneration in iKO mice. (Fig. 3l-q and Fig. S3c-e). The above findings thus strengthened our claim that ATF3 prevents SC activation. The newly added results can be found on Page 10 of the revised manuscript. We also over-expressed H2B in vitro and in vivo. In vitro we found that over-expressing H2B by a transfecting a pcDNA-H2B plasmid in FISCs indeed repressed accelerated activation of Atf3 iKO cells but had no impact on Ctrl cells (Fig. 7a-b). And in vivo, we also over-expressed H2B by intramuscular injection of H2B expressing lentivirus and found that H2B overexpression indeed blunted the accelerated regeneration in Atf3-iKO mice (Fig. 7c-g). Altogether these findings validate that H2B loss indeed mediates the precocious SC activation and enhanced muscle regeneration. The newly added results can be found on Page 15 of the revised manuscript.

3.10 P13: *“These results demonstrate that the ATF3 loss decreases H2b gene expression”. This is counter-intuitive as Atf3 KO leads to activation and proliferation, which require generation of new histone protein.*

A: Thanks for the critical question. As answered in comment 3.5, it is likely the reduced amount of H2B may lead to formation of non-canonical nucleosomes, for example, so called half-nucleosomes consisting of one copy of each of the four core histones or hexasomes with two copies of H3/H4 and one copy of H2A/H2B. The existence of half-nucleosomes in fact facilitates the transcription of activation/differentiation genes thus promotes SC activation.

3.11 *Full name of ATF3 should be provided at the first occurrence. ATF3 is known to activate gene expression and it is puzzling why most genes are upregulated in Atf3 KO satellite cells. Some discussion might be helpful.*

A: Thanks for the great suggestion. We have now added the full name on page 2 of the revised text. ATF3 is known to play both activating and repressing functions in cell dependent manner. ATF3 homodimers and heterodimers (with other bZip proteins) repress and induce gene expression, respectively²³. For example, ATF3 can repress the expression of proinflammatory cytokines induced by the toll-like receptor 4 in the immune response²⁴. ATF3 is induced during the early stage of paligenosis to transcriptionally activate the lysosomal trafficking gene Rab7b²⁵. We have added the above discussion on page 4 of the revised text.

3.12 Abstract: “which reduction accelerates nucleosome displacement and gene transcription required for SC activation”. This sentence appears to have grammar issue. Change to “whose reduction”

A: Thanks for the comment and we apologize for the mistake. We have made the change on page 2 of the revised abstract.

3.13 *The Atf3 KO phenotype is very similar to, but milder than, Pten KO phenotypes (PMID: 27880908, PMID: 28094257). In discussion, the authors discussed other quiescent regulators and PTEN should be included. Also, there are reports that ATF3 and PTEN may phenocopy and regulate each other (PMID: 25531328, PMID: 27308526). These should also be discussed.*

A: Thanks for the great suggestion. We have now included Pten in the list of SC quiescence maintenance factors on Page 19 of the revised Discussion. Nevertheless, we feel elaborated discussion on PTEN and ATF3 connection may break the logical flow of the context.

1. Marzluff WF, Gongidi P, Woods KR, Jin J, Maltais LJ. The Human and Mouse Replication-Dependent Histone Genes. *Genomics*. 2002/11/01/ 2002;80(5):487-498.
2. Almada AE, Wagers AJ. Molecular circuitry of stem cell fate in skeletal muscle regeneration, ageing and disease. *Nat Rev Mol Cell Biol*. May 2016;17(5):267-279.
3. Machado L, Esteves de Lima J, Fabre O, et al. In Situ Fixation Redefines Quiescence and Early Activation of Skeletal Muscle Stem Cells. *Cell Rep*. Nov 14 2017;21(7):1982-1993.
4. van Velthoven CTJ, de Morree A, Egner IM, Brett JO, Rando TA. Transcriptional Profiling of Quiescent Muscle Stem Cells In Vivo. *Cell Rep*. Nov 14 2017;21(7):1994-2004.
5. van den Brink SC, Sage F, Vértesy Á, et al. Single-cell sequencing reveals dissociation-induced gene expression in tissue subpopulations. *Nat Methods*. Sep 29 2017;14(10):935-936.
6. Almada AE, Horwitz N, Price FD, et al. FOS licenses early events in stem cell activation driving skeletal muscle regeneration. *Cell Rep*. Jan 26 2021;34(4):108656.
7. Barutcu AR, Elizalde G, Gonzalez AE, et al. Prolonged FOS activity disrupts a global myogenic transcriptional program by altering 3D chromatin architecture in primary muscle progenitor cells. *Skeletal Muscle*. 2022/08/15 2022;12(1):20.

8. Liu Y, Chen Y, Deng X, Zhou J. ATF3 Prevents Stress-Induced Hematopoietic Stem Cell Exhaustion. *Frontiers in cell and developmental biology*. 2020;8:585771.
9. Tsujino H, Kondo E, Fukuoka T, et al. Activating transcription factor 3 (ATF3) induction by axotomy in sensory and motoneurons: A novel neuronal marker of nerve injury. *Mol Cell Neurosci*. Feb 2000;15(2):170-182.
10. Wanner R, Knöll B. Interference with SRF expression in skeletal muscles reduces peripheral nerve regeneration in mice. *Sci Rep*. Mar 24 2020;10(1):5281.
11. Cisterna B, Giagnacovo M, Costanzo M, et al. Adapted physical exercise enhances activation and differentiation potential of satellite cells in the skeletal muscle of old mice. *Journal of anatomy*. May 2016;228(5):771-783.
12. Fry CS, Noehren B, Mula J, et al. Fibre type-specific satellite cell response to aerobic training in sedentary adults. *The Journal of physiology*. Jun 15 2014;592(12):2625-2635.
13. Macaluso F, Brooks NE, van de Vyver M, Van Tubbergh K, Niesler CU, Myburgh KH. Satellite cell count, VO₂(max) , and p38 MAPK in inactive to moderately active young men. *Scandinavian journal of medicine & science in sports*. Aug 2012;22(4):e38-44.
14. Joannis S, McKay BR, Nederveen JP, et al. Satellite cell activity, without expansion, after nonhypertrophic stimuli. *American journal of physiology. Regulatory, integrative and comparative physiology*. Nov 1 2015;309(9):R1101-1111.
15. Mackey AL, Karlsen A, Couppé C, et al. Differential satellite cell density of type I and II fibres with lifelong endurance running in old men. *Acta physiologica (Oxford, England)*. Mar 2014;210(3):612-627.
16. He L, Ding Y, Zhao Y, et al. CRISPR/Cas9/AAV9-mediated in vivo editing identifies MYC regulation of 3D genome in skeletal muscle stem cell. *Stem Cell Reports*. Oct 12 2021;16(10):2442-2458.
17. Chen X, Xue G, Zhao J, et al. Lockd promotes myoblast proliferation and muscle regeneration via binding with DHX36 to facilitate 5' UTR rG4 unwinding and Anp32e translation. *Cell Reports*. 2022/06/07/ 2022;39(10):110927.
18. So KKH, Huang Y, Zhang S, et al. seRNA PAM controls skeletal muscle satellite cell proliferation and aging through trans regulation of Timp2 expression synergistically with Ddx5. *Aging Cell*. 2022;21(8):e13673.
19. Zhao Y, Ding Y, He L, et al. Multiscale 3D genome reorganization during skeletal muscle stem cell lineage progression and aging. *Science Advances*. 2023;9(7):eab01360.
20. Qiao Y, Sun Q, Chen X, et al. Nuclear m6A reader YTHDC1 promotes muscle stem cell activation/proliferation by regulating mRNA splicing and nuclear export. *eLife*. 2023/03/09 2023;12:e82703.
21. Proske U, Morgan DL. Muscle damage from eccentric exercise: mechanism, mechanical signs, adaptation and clinical applications. *J Physiol*. Dec 1 2001;537(Pt 2):333-345.
22. Sousa-Victor P, García-Prat L, Muñoz-Cánoves P. Control of satellite cell function in muscle regeneration and its disruption in ageing. *Nature Reviews Molecular Cell Biology*. 2022/03/01 2022;23(3):204-226.
23. Hai T, Wolfgang CD, Marsee DK, Allen AE, Sivaprasad U. ATF3 and stress responses. *Gene Expr*. 1999;7(4-6):321-335.
24. Gilchrist M, Thorsson V, Li B, et al. Systems biology approaches identify ATF3 as a negative regulator of Toll-like receptor 4. *Nature*. May 11 2006;441(7090):173-178.

25. Radyk MD, Spatz LB, Peña BL, et al. ATF3 induces RAB7 to govern autodegradation in paligenosis, a conserved cell plasticity program. *EMBO Rep.* Sep 6 2021;22(9):e51806.

REVIEWERS' COMMENTS

Reviewer #1 (Remarks to the Author):

Cut & Run experiments have been improved as requested.

I have no further questions.

Reviewer #2 (Remarks to the Author):

The authors have made a strong effort to address all of the points raised in my initial review. Their manuscript is acceptable for your journal.

Reviewer #3 (Remarks to the Author):

The authors did a great job addressing my concerns